# Condensate droplet roaming on nanostructured superhydrophobic surfaces

Cheuk Wing Edmond Lam [1,3], Kartik Regulagadda [1,4], Matteo Donati [1], Abinash Tripathy[1], Gopal Chandra Pal[2], Chander Shekhar Sharma [2], Athanasios Milionis[1] & Dimos Poulikakos [1] ✉

Jumping of coalescing condensate droplets from superhydrophobic surfaces is an interesting phenomenon which yields marked heat transfer enhancement over the more explored gravity-driven droplet removal mode in surface condensation, a phase change process of central interest to applications ranging from energy to water harvesting. However, when condensate microdroplets coalesce, they can also spontaneously propel themselves omnidirectionally on the surface independent of gravity and grow by feeding from droplets they sweep along the way. Here we observe and explain the physics behind this phenomenon of roaming of coalescing condensate microdroplets on solely nanostructured superhydrophobic surfaces, where the microdroplets are orders of magnitude larger than the underlaying surface nanotexture. We quantify and show that it is the inherent asymmetries in droplet adhesion during condensation, arising from the stochastic nature of nucleation within the nanostructures, that generates the tangential momentum driving the roaming motion. Subsequent dewetting during this conversion initiates a vivid roaming and successive coalescence process, preventing condensate flooding of the surface, and enhancing surface renewal. Finally, we show that the more efficient conversion process of roaming from excess surface energy to kinetic energy results in significantly improved heat transfer efficiency over condensate droplet jumping, the mechanism currently understood as maximum.

The phenomenon of water vapour condensation on a surface begins with the formation of discrete liquid nuclei, which grow into droplets that can coalesce with one another. If not removed periodically, such droplets of condensate form a thick continuous film, which hinders the removal of heat from the vapour through the surface. The pursuit for efficient heat removal has strongly motivated surface engineering research, with the central idea to minimise the residence time and amount of the liquid condensate on the cooled surface[1–10].

On superhydrophobic surfaces, it is possible for condensate microdroplets to spontaneously depart in the direction normal to the surface, by jumping upon coalescence, converting released surface energy to kinetic energy[11,12]. Such jumping droplet departure significantly reduces the size of droplets residing on the surface, further improving heat transfer efficiency compared to conventional dropwise condensation[13,14]. On superhydrophobic surfaces with microstructures, this spontaneous motion can also be at an angle, or even tangential, instead of normal to the surface[15–23]. It has been postulated that[15,17–19,24], as the condensate droplets are at the same length scale as the individual microfeatures, coalescence on the side walls of the microstructure cavities triggers inclined jumping in random

[1]Laboratory of Thermodynamics in Emerging Technologies, Department of Mechanical and Process Engineering, ETH Zurich, Zurich, Switzerland. [2]Thermofluidics Research Laboratory, Department of Mechanical Engineering, Indian Institute of Technology Ropar, Rupnagar, Punjab, India. [3]Present address: Department of Mechanical Engineering, Massachusetts Institute of Technology, Cambridge, MA, USA. [4]Present address: Department of Mechanical Engineering, UC Berkeley, Berkeley, CA, USA. ✉e-mail: dpoulikakos@ethz.ch

directions. However, tangential movement of coalescing condensate microdroplets is also observed on superhydrophobic surfaces equipped solely with nanostructures[25,26], which are orders of magnitude smaller than the droplets in concern, with remarkable effects on ensuing heat transfer augmentation. Here we explain this unexplored droplet roaming coalescence mechanism, identify the conditions under which roaming occurs, and determine its significant effect on heat removal from a surface.

As in-plane roaming can span great lengths and coalesce with other condensate microdroplets along the way[25], compared to out-of-plane jumping which is confined to a local cluster of droplets, it provides a pathway to continuously remove larger amounts of condensate[15]. Frequent roaming exposes needed underlying surface for new nucleation cycles, thus reducing droplet sizes against conventional gravity-driven dropwise condensation, and ultimately markedly improving heat transfer, even compared to jumping dropwise condensation.

## Results and discussion
### Roaming on solely nanostructured surfaces
To allow high-speed imaging of roaming, we prepare a reflective solely nanostructured superhydrophobic surface by exposing flat aluminium substrates to hot water to form boehmite nanowalls[27–29], which are

then coated with hydrophobic poly-(1H,1H,2H,2H-perfluorodecyl acrylate) (pPFDA) using initiated chemical vapour deposition (iCVD)[2,15] (Methods). The coating conforms to the nanowalls, and its thickness is measured to be 3.5 nm with ellipsometry (Supplementary Information S1). An image of the nanostructures with the pPFDA coating using scanning electron microscopy (SEM) is seen in Fig. 1a. The advancing contact angle and contact angle hysteresis are 162.5° ± 1.8° and 1.1°, respectively (Methods).

All samples are tested in our condensation setup illustrated in Fig. 1b. A transparent window separates the condensation chamber environment and the atmosphere. During experiment, saturated steam (30 mbar, 24.1 °C) continuously passes over and condenses on the cooled surface of the sample. A microscope objective in front of the window enables direct observation of microscale condensation behaviour with a high-speed camera at a resolution of 4.5 μm per pixel, and temperature sensors in the chamber allow the simultaneous measurement of heat transfer performance. See Supplementary Information S2.

We investigate roaming motion on solely nanostructured superhydrophobic surfaces and avoid the presence of microstructures which can alter the motion of coalescing condensate microdroplets at the similar length scale (Supplementary Information S3). A typical roaming event is shown in Fig. 1c. After the first coalescence (Panel i),

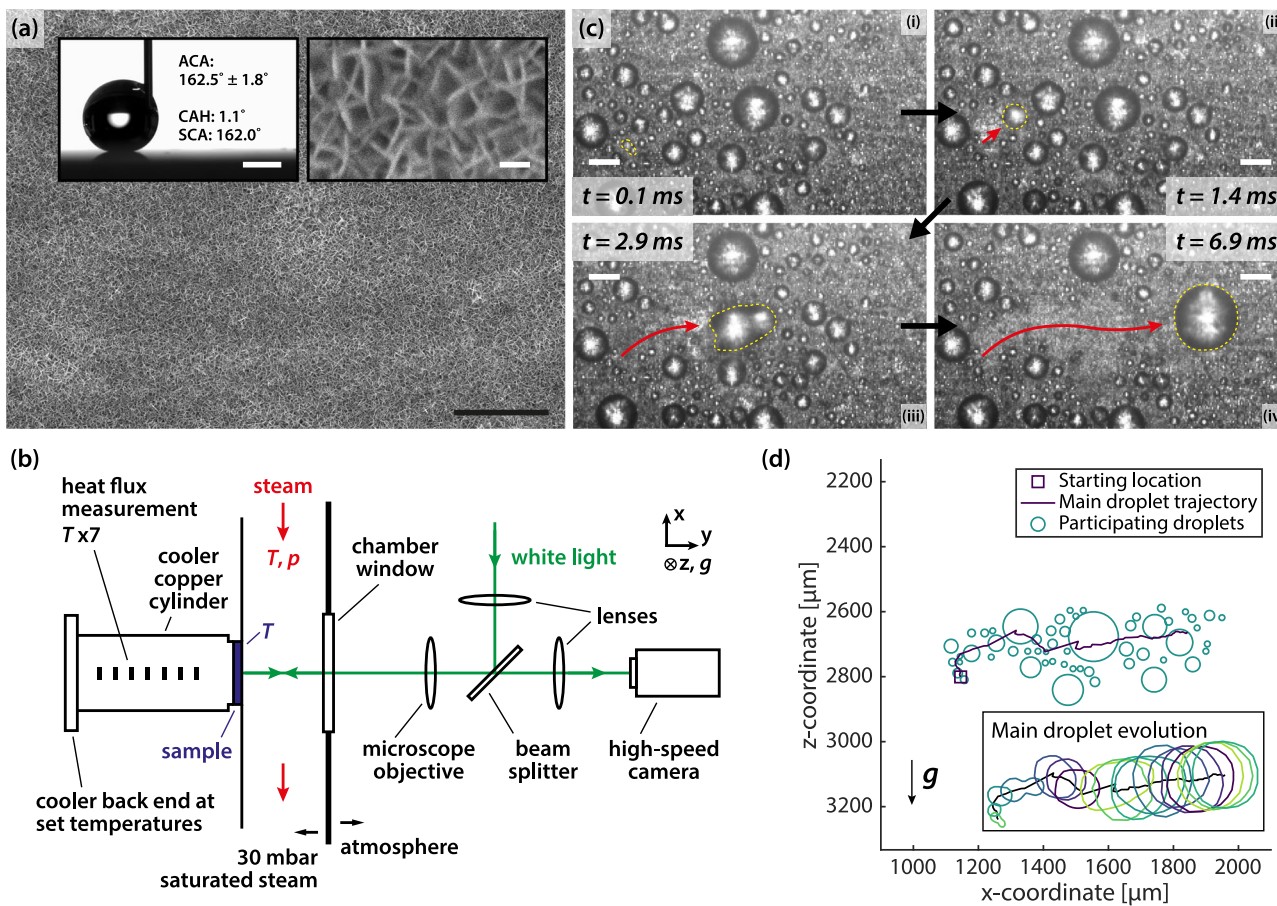

**Fig. 1 | Roaming on solely nanostructured superhydrophobic surfaces. a** SEM image of the boehmite nanowalls coated with pPFDA, a solely nanostructured superhydrophobic surface. Scale bar: 2 μm. Left inset: Water droplet being deposited at 2 μL s⁻¹, and wettability measurements of the advancing contact angle (ACA), contact angle hysteresis (CAH), and the static contact angle (SCA). Scale bar: 1 mm. Right inset: SEM image of the nanowalls at higher magnification. Scale bar: 100 nm. **b** Schematic of the condensation and observation setup. *T* and *p* refer to temperature and pressure measurements respectively. Gravity *g* is in the +*z*-direction.

**c** Roaming event during vapour condensation on boehmite nanowalls coated with pPFDA. Yellow dashed lines enclose the main droplet. Red arrow indicates the approximate trajectory of the roaming event. Also see Supplementary Movie 1. Subcooling: 2.6 K. Gravity is downwards. Scale bars: 100 μm. **d** Participating droplets distribution for the in-plane (xz) roaming event in (**c**). The line represents the trajectory of the main droplet (red arrow in **c**). Inset: Evolution of the shape of the main droplet. Every contour is 0.4 ms apart. Source data are provided as a Source Data file.

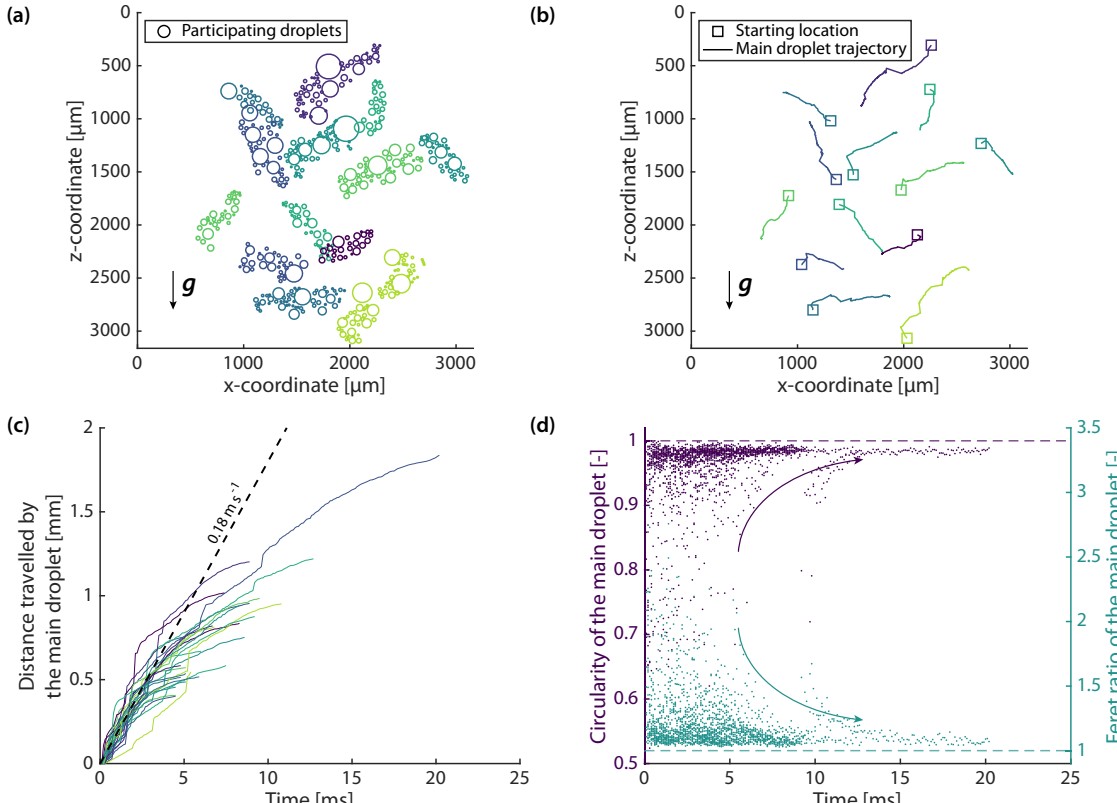

**Fig. 2 | Characteristics of roaming events. a** Participating droplet distribution of 13 selected roaming events (indicated by different colours), at different locations of the surface. Roaming events do not repeatedly occur at the same location over time. **b** Main droplet trajectories corresponding to the events shown in (**a**). Squares indicate starting location of the events. All events progress in in-plane directions, independent of downward gravity. **c** Distance travelled by the main droplet for all the events. Initial roaming velocity $0.18\,\mathrm{m\,s^{-1}}$. **d** Circularity ($=4\pi(\mathrm{area}/\mathrm{perimeter}^2)$) and Feret ratio ($=\mathrm{maximum\,caliper\,diameter}/\mathrm{minimum\,caliper\,diameter}$) of the main droplet from the events. Both approach unity at $\approx 5\,\mathrm{ms}$. Source data are provided as a Source Data file.

there is a tangential motion to the right (Panel ii). The droplet coalesces with other droplets along its way, roaming the surface (Panel iii), before coming to rest (Panel iv). The corresponding video can be found in Supplementary Movie 1. Evidently, roaming demonstrates in-plane arbitrary directionality which spans across considerable time and distance, and is distinct from localised multi-droplet coalescence[30,31], which is confined to a specific location and occurs on shorter timescales (Supplementary Movie 2).

We observe the condensation behaviour as we increase the surface subcooling $\Delta T = T_{\mathrm{steam}} - T_{\mathrm{surf}}$, the difference between the steam temperature $T_{\mathrm{steam}}$ and the surface temperature $T_{\mathrm{surf}}$. As the steam is always at saturation, $T_{\mathrm{steam}}$ equals the saturation temperature at our operating pressure ($T_{\mathrm{sat}}$ at 30 mbar). Above a subcooling of $\approx 1.5\,\mathrm{K}$, roaming becomes frequent on the surface. We characterise roaming events from two perspectives: (1) We measure the position and time of all visible droplets which coalesce and take part in the event (participating droplets, shown as circles in Fig. 1d); and (2) we track over time the location and shape of the travelling droplet (main droplet, trajectory and shape shown in inset of Fig. 1d) which grows as it coalesces with and absorbs the participating droplets. Details of image processing, and individual droplet measurement and tracking can be found in Supplementary Information S4.

### Mechanism of roaming coalescence

Figure 2a, b displays the participating droplet distributions and the corresponding trajectory of the main droplet of several roaming events, out of a total of 28 observed at a subcooling of 2.2–2.6 K. Roaming occurs in all in-plane directions, independent of gravity. This can be explained with the Bond number $\mathrm{Bo} = \Delta\rho g R^2/\sigma$, where $\Delta\rho$ is the density difference of liquid water and its saturated vapour, $g$ is the gravitational acceleration, $R$ is the characteristic droplet length scale, and $\sigma$ is the surface tension of liquid water. Substituting the mean main droplet equivalent radius of $79 \pm 28\,\mu\mathrm{m}$ as the length scale, $\mathrm{Bo} = 0.0009 \ll 1$, indicating that roaming is dominantly a capillary phenomenon. During our experiments, we have not seen roaming events to repeatedly occur at fixed locations on the condensing surface (i.e. the starting and terminating locations are random, see Fig. 2b), indicating that it is not triggered by surface defects. Roaming is also found to be independent of the form of nanostructures, as we have observed its occurrence on titanium dioxide nanorods[32] and copper(II) hydroxide nanoneedles[33] as well (Supplementary Information S5). We have not found a specific nanomorphology which would prohibit roaming of condensate droplets, and we believe that this phenomenon is generic to nanostructured superhydrophobic surfaces. Occasionally, a roaming event can alter its direction as it progresses, resulting in an in-plane curved trajectory (Supplementary Movie 3). Apart from coming to rest as in Fig. 1c, it can also terminate by jumping (Supplementary Movie 4). Of the measured roaming events, the mean duration is $5.3 \pm 3.4\,\mathrm{ms}$, with a mean travelled distance of $744 \pm 334\,\mu\mathrm{m}$, on average 37× of the mean participating droplet diameter of the event.

Roaming requires significant generation of tangential momentum. Figure 2c displays the distance travelled over time for the main droplet of roaming events. Although the mass increase of the main droplet varies significantly for different events (Supplementary Information S6), it largely follows a constant initial velocity of $\approx 0.18\,\mathrm{m\,s^{-1}}$,

before slowing and diverging at ~3–5 ms as viscous dissipation sets in (compared to a viscous timescale $t_\mu = \rho R^2/\mu = 7$ ms, where $R = 79$ μm is the mean main droplet equivalent radius and $\mu$ is the dynamic viscosity of liquid water). We term this velocity of the travelling main droplet as the apparent roaming velocity. Although roaming occurs in all in-plane directions, including some against gravity, there is no visible effect on the velocity, further indicating that it is dominated by capillarity effects.

As the main droplet gains mass and size, perturbations from further coalescence with upcoming participating droplets increasingly contribute to the low-amplitude capillary waves at the liquid-vapour interface, instead of bulk droplet motion. This is due to the increase in the number of available oscillation modes in a larger main droplet[34,35], along with coalescence bridges becoming increasingly small compared to the traversing main droplet. As viscous effects become important, the roaming event slows and terminates. We quantify the intensity of coalescence over the course of the multiple events by describing the shape evolution of the main droplet in Fig. 2d. At the transition time of $\approx 5$ ms, its circularity and Feret ratio quickly approach unity, indicating transition to a circular contour (also see inset of Fig. 1d). At the same time, the ratio of participating droplet sizes relative to the main droplet size drops below unity (Supplementary Information S6).

The translational kinetic energy of roaming stems from the excess surface energy due to the reduction of liquid-vapour interfacial area upon coalescence. Instead of the absolute velocity, we compare and normalise the roaming velocity with the theoretical maximum velocity (i.e. if all excess surface energy were converted to in-plane translational kinetic energy) to quantify the efficiency of this energy conversion. We employ initial velocities just after coalescence, as common in the literature of droplet-jumping studies, to exclude effects other than capillarity. However, we note that even without tangential momentum generation, there would be a shift in the location of the main droplet after it coalesces with each participating droplet, due to the addition of mass from the participating droplet to the main droplet away from its location. Therefore, to account for this effect and extract the "real" roaming velocity that is purely the result of tangential momentum generation, we dynamically measure the increase in the distance between the main droplet and the centre of mass of the system of coalesced participating droplets. The real roaming velocity is $48 \pm 14\%$ of the theoretical maximum. For out-of-plane droplet jumping, it is $\approx 20\%$[11,36]. See Supplementary Information S7. Roaming better scavenges the excess surface energy of coalescence, which would otherwise be dissipated as heat, for condensate removal, improving heat transfer efficiency.

### Roaming condensation heat transfer and the transition subcooling

The more efficient energy conversion and larger span of roaming than jumping suggest heat transfer benefits. In Fig. 3, the heat transfer performance and the condensation behaviour at different subcooling are quantified. We first measure the heat flux $q''$ and the subcooling $\Delta T$ (Supplementary Information S8) and compute the heat transfer coefficient $h = q''/\Delta T$ (Fig. 3a). A surface with the same boehmite nanowalls but without the pPFDA coating (pristine boehmite) is used as the filmwise condensation control. Filmwise condensation measurements are validated against the Nusselt model, as commonly found in literature[37] (Supplementary Information S2).

Overall, the superhydrophobic boehmite surface is superior: When comparing the mean of all measurements from each surface, there is an increase in the heat transfer coefficient of over 300% from 20.1 kW m$^{-2}$ K$^{-1}$ on pristine boehmite to 82.6 kW m$^{-2}$ K$^{-1}$ on the superhydrophobic surface. However, on the superhydrophobic surface alone, there are two regimes of condensation mode, dependent on the current subcooling. At low subcooling, condensation is dominated by

the jumping dropwise mode with a relatively lower heat transfer coefficient (mean = 62.7 kW m$^{-2}$ K$^{-1}$, first 3 points from the left in Fig. 3a); but when subcooling increases, there is a transition and condensation is dominated by the roaming mode with an increased heat transfer coefficient (mean = 97.5 kW m$^{-2}$ K$^{-1}$, last 4 points from the left in Fig. 3a). When compared to filmwise condensation at similar subcooling (mean = 25.4 kW m$^{-2}$ K$^{-1}$, first two points from the left in Fig. 3a), jumping dropwise condensation provides a 147% increase in the heat transfer coefficient while roaming condensation provides a 284% increase. The synergistic effect of a higher heat transfer coefficient at a higher thermal driving force, i.e. subcooling, results in a 175% higher heat flux for roaming condensation than jumping dropwise condensation. The jumping-roaming transition can be seen in Fig. 3b and Supplementary Movie 5. We quantify the transition in the top subplot of Fig. 3c and show that when the subcooling increases past the transition at $\approx 1.5$ K, the surface area renewal rate $S'$ from roaming sharply increases. Remarkably, over 70% of the surface is renewed every second by roaming when it is the dominant mode. Lastly, most of the roaming events end as droplets at rest. Termination in jumping is scarce when roaming first emerges at the transition subcooling; as subcooling increases, droplet jumping of all kind all but vanishes. See Supplementary Movie 5 and Supplementary Information S9. A few rare cases of jumping at subcooling higher than the transition can be seen in Supplementary Movie 2. The gradual deceleration of a roaming droplet until coming to rest (instead of an abrupt stop) on the surface, as evident in Fig. 2c, indicates that motion is not terminated by pinning at local adhesion points from condensate-filled nanostructures, but rather by energy dissipation over time. A droplet coming to rest on the surface after roaming is therefore likely in the Cassie state and relatively mobile. This larger droplet is then available for further roaming and coalescence. The entire process significantly speeds up the growth of droplets on the surface to attain the gravitational departure size (Supplementary Information S9). Eventually, the condensate droplets leave by gravity.

The roaming mode provides higher heat transfer efficiency than the jumping dropwise mode despite a surface with more larger droplets as seen in Fig. 3b. In condensation, most of the heat and mass transfer is attributed to the initial droplet growth after nucleation[38]. On a surface with a distribution of various condensate droplet sizes, the majority of heat flows through the smallest droplets. When subcooling increases, (1) the diameter at which nucleation occurs, i.e. the critical nucleation diameter $d_{crit}$, decreases, and (2) the nucleation rate (number of nuclei per area per time) increases[39]. Therefore, at elevated subcooling, a renewed surface area is soon filled with a large number of small condensate droplets ideal for heat transfer. The frequent renewal of large surface areas (1) by roaming itself, and (2) from the increased gravitational departure assisted by roaming, enable abundant renucleation and ultimately maximise heat transfer. In addition, the higher condensation rates when subcooling increases, inevitably translate to more active condensation behaviour. For example, there is a higher droplet density and thus a higher frequency of (roaming or localised) coalescence events. Such collateral effects, together with the prominent roaming motion of droplets over the entire surface, piece together the overall roaming condensation mode with its high heat transfer coefficients.

The emergence of roaming when subcooling increases provides a clue about its origin. As $d_{crit}$ is reduced with increasing subcooling and becomes smaller than the nanocavity sizes, nucleation occurs stochastically within the nanocavities. At the transition subcooling of 1.5 K, $d_{crit}$ ($\approx 23$ nm) is below the majority of boehmite cavity sizes (Fig. 3c). On the other hand, we find that on surfaces with much sparser nanostructures and thus larger cavities such as copper(II) hydroxide nanoneedles, the transition subcooling is notably reduced to 0.7 K (Supplementary Information S10). In addition, at a subcooling of 1.3 K, these sparser copper(II) hydroxide structures begin to flood as most of

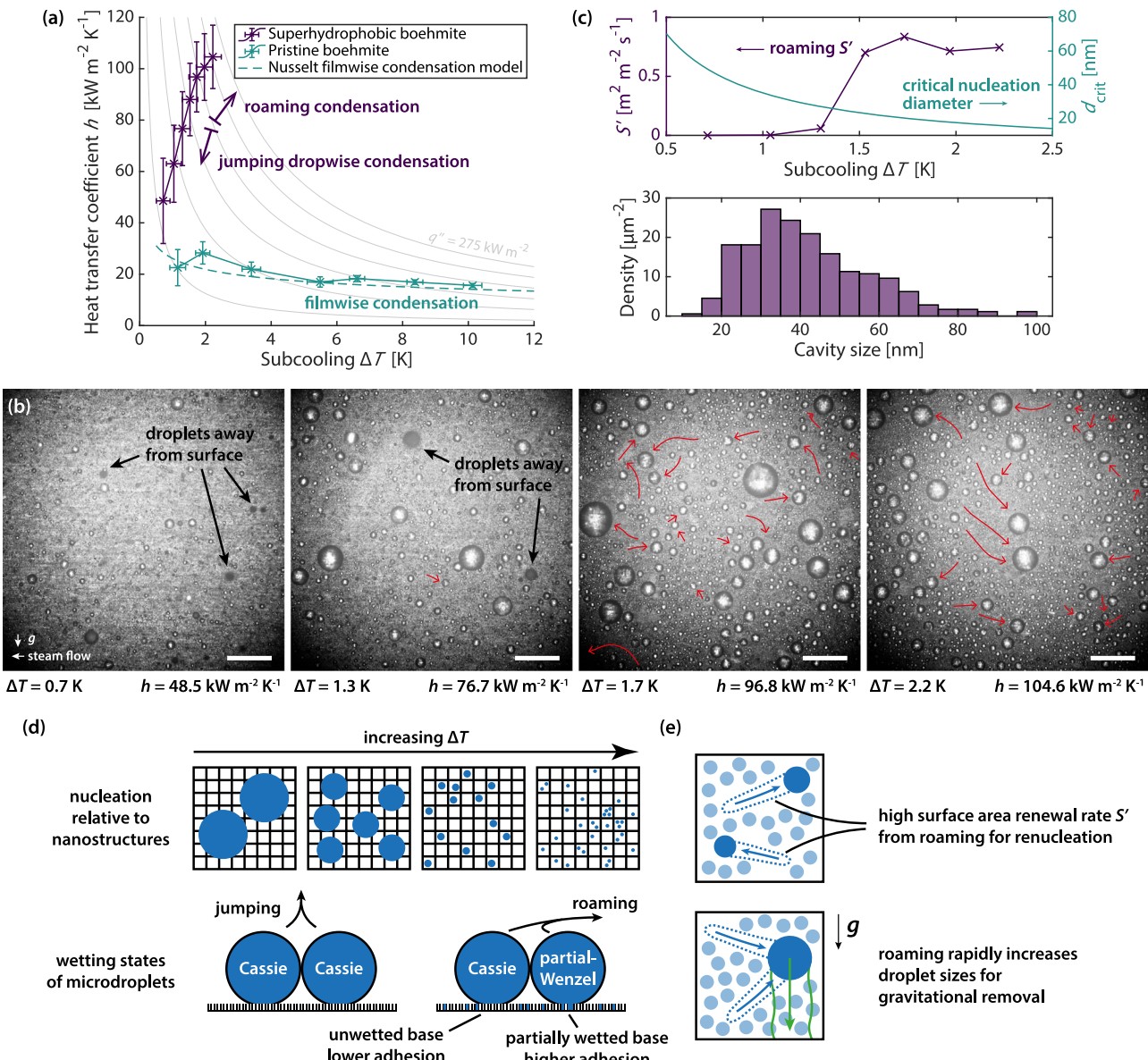

**Fig. 3 | Heat transfer performance of roaming condensation. a** Heat transfer coefficients $h$ at steady state. Lines of constant heat flux $q''$ are shown in grey, from 25 to 275 kW m$^{-2}$ at intervals of 50. For a fair test, the 7 subcooling achieved for each surface correspond to 7 identical cooler back end temperatures (Supplementary Information S2). On the superhydrophobic surface, two modes of condensation are observed. Measurements on pristine boehmite match closely with the Nusselt model for filmwise condensation. **b** Snapshots of condensation behaviour for superhydrophobic boehmite. Transition is seen from jumping dropwise to roaming condensation. At the lowest subcooling (0.7 K), only jumping is observed and there are numerous droplets in the vapour, with darker appearance and out-of-focus contour. These droplets in the vapour travel in one general direction to the bottom left due to steam flow (leftward) and gravity (downward). At 1.3 K, the number of jumped droplets in the vapour is visibly reduced, and some are seen to return to the surface. After the transition subcooling (1.5 K), condensation is dominated by

roaming. Red arrows are trajectories of roaming events. Roaming droplets travel in all in-plane directions. See Supplementary Movie 5 for the corresponding video. Scale bars: 500 μm. **c** Surface area renewal rate $S'$ from roaming (unit: m$^2$ of surface area renewed per m$^2$ of condensing surface per second) and critical nucleation diameter $d_{crit}$ for 30 mbar saturated steam (top). As transition to roaming occurs at ≈1.5 K, the critical nucleation diameter lies below most nanostructure cavity sizes (bottom). The sizes are obtained from the square root of the projected area of each cavity (Supplementary Information S11). **d** When subcooling is increased past the transition, condensate nucleates within the nanostructures. Microdroplets on top of these nanostructures could then exhibit different wetting states. The asymmetric adhesion gives rise to substantial tangential momentum upon coalescence. **e** The high surface area renewal rate of roaming enables abundant renucleation. Frequent roaming also assists droplet growth to the required size of gravitational removal. Source data are provided as a Source Data file.

the nanocavities are filled with condensate. This jumping-roaming-flooding transition evidently shows that roaming is closely related to condensate filling of some nanocavities, producing some droplets on the nanostructures which are in the partial-Wenzel state. It furthers droplet growth and heat transfer as there is more droplet-substrate contact than Cassie-state droplets[40]. Figure 3d, e summarises the mechanism for roaming and its benefits to heat transfer, which we will discuss in detail in the upcoming sections.

For a cavity to be filled, two competing factors are in play. When subcooling increases, although the nucleation rate increases, the volume of each nucleus reduces. To assess the probability of cavity filling through nucleation, we define a volumetric nucleation rate as the product of the two. A substantial increase in the volumetric nucleation rate is seen around the transition subcooling at 1.5 K, indicating the increased likelihood for the cavities to be filled. The timescale to fill the nanocavities

of boehmite is in turn estimated to be 0.1–1 ms. See Supplementary Information S11.

As roaming is not observed to repeatedly initiate at certain locations (Fig. 2b), and the surface can sustain roaming condensation at steady state with no surface flooding over time, such stochastic locally wetted cavities are expected to dewet in a roaming event, similar to dewetting by condensate motion previously suggested[11] for the case of droplet jumping. On the other hand, hierarchical condensation[41], in which condensate nucleates within structure cavities under a droplet, is very unlikely, due to the lack of low-resistance vapour flow paths across the thin, dense and closed nanostructure (Supplementary Information S3). After roaming renews the surface, the dewetted cavities are again exposed to the vapour for another nucleation cycle.

The heat transfer coefficient can exhibit different trends with subcooling for surfaces on which condensate appears in the form of droplets[4,13,42,43]. When droplet removal is sufficiently efficient, it can increase with subcooling[4,42], due to higher nucleation rates and smaller nucleation diameters. This important trend is also present in the superhydrophobic boehmite heat transfer coefficients in Fig. 3a. However, this trend sustains so long as the condensation mode allows it, that is, there is limited flooding of structures or saturation of nucleation sites because of the efficient droplet removal. The >70% area renewed every second in roaming condensation enables ample space for fresh nucleation, sustaining the trend. For structured surfaces, when the subcooling is high enough for nucleation to occur within the cavities, apart from the increased nucleation rate from the increase in subcooling, there is as well additional area available for nucleation from the cavity walls. These higher rates are only sustainable when they are balanced with enhanced condensate removal from the cavities to avoid flooding. Surface structures which promote the ejection of droplets from within are often exploited to achieve this goal[44–48], and delay flooding so that the abovementioned relationship can be extended to higher subcooling levels for jumping dropwise condensation. In our work, we show that as jumping subsides at higher subcooling, roaming, which naturally occurs on these superhydrophobic surfaces, can further extend the trend at high subcooling, by providing an efficient pathway to employ the excess surface energy of droplet coalescence for cavity dewetting. See Supplementary Information S8 for additional discussion.

In summary, we have put forth roaming condensation as a readily achievable state for a wide range of superhydrophobic surfaces. It is a condensation mode in itself, and it occurs sustainably in the requisite subcooling range, as a result of the collaborative effect of nucleation diameters and nucleation rates. The lack of flooding, and the highest heat transfer coefficients roaming yields compared to all other modes, including jumping dropwise condensation, render it as a preferable mode to be targeted in various heat transfer applications for maximum energy efficiency.

## Generation of tangential momentum in roaming

The absence of roaming at the limit of low subcooling suggests that this phenomenon is exclusive to condensate droplets—gently deposited Cassie droplets do not roam upon coalescence. Indeed, no tangential motion has been reported in the literature for deposited droplets on solely nanostructured surfaces. As some nanocavities fill, stochastic wetting at random locations across the surface promotes the concurrent presence of Cassie and (partial-)Wenzel-state condensate droplets on superhydrophobic surfaces[40]. Their different adhesion[47] produces an adhesion asymmetry of the droplets for tangential motion. See Fig. 3d. Without this, there is no apparent reason for out-of-plane jumping to transition to in-plane roaming. Jumping is a result of the reaction force from a rapidly growing droplet coalescence bridge impinging on the surface, breaking the oscillation symmetry[49,50]. The normal impingement of the bridge can only generate a normal reaction from the surface in the opposite direction. For

roaming, there is no symmetry-breaking surface for a tangential reaction force to manifest as the condensate microdroplets are orders of magnitude larger than the underlaying nanostructures. If the increase in adhesion for all condensate droplets on the surface were uniform when subcooling increases, i.e. no asymmetry, jumping would gradually cease and transition to flooded condensation would be directly evident without any intermediate in-plane roaming regime. The excess surface energy from coalescence would no longer overcome the increased adhesion and be dissipated instead. Droplets would not depart at all, whether in-plane or out-of-plane, until they attain the size when gravity dominates.

Moreover, any droplet size mismatch during coalescence cannot explain the generation of tangential momentum as well. In our roaming events, there is no observable trend in participating droplet sizes (Fig. 2a). Additionally, for two size-mismatched coalescing droplets, the reaction force from the symmetry-breaking surface would still largely be normal to the surface. Numerical simulations (Methods) confirm that the direction of jumping from the coalescence of two size-mismatched droplets deviates <4° from the surface normal, in line with previously reported results[24]. See Supplementary Information S12.

While nuclei within the nanostructure cavities may coalesce with and effectively be absorbed by the microdroplets on top, we do not expect any substantial motion of the microdroplet to result from such coalescence. It is because the vast difference in size of the coalescing droplets quickly dampens capillary waves before they are converted to bulk droplet motion. Moreover, when subcooling increases, this becomes increasingly difficult and improbable due to the high nucleation rates within the structure cavities. Microdroplets with different wetting states will eventually form. Roaming is a consequence of the adhesion asymmetry of droplets under different wetting states and propagates by the dewetting of the partial-Wenzel state droplets.

Roaming condensation is stochastic and occurs on very actively condensing surfaces under saturated steam, where isolation and control of individual events are impossible. Current experimental methods do not simultaneously possess sufficient spatial and temporal resolution to visualise the pinned contact line of the adhered droplet, or the ≲100 nm-thick wetted nanostructure layer below the droplet, during coalescence. Therefore, to further support our conclusions from the experimental findings above, we follow with numerical simulations, which yield essential additional information on the coalescence mechanism. Guided by the experiments, we set up simulation cases to study the evolution of momentum and energies throughout the process in a highly temporally resolved manner, which cannot be obtained experimentally, to demonstrate motion resulting from the wetting asymmetry of two participating droplets. See Fig. 4a and Methods. Two equally sized droplets of 160 μm in diameter are first placed on a substrate at a contact angle of 160°, and a symmetry plane is specified at $z = 0$. To mimic the effect of a wetted nanostructure layer below a droplet (D1 in Fig. 4a), we specify the contact angle only for its base area as 2°.

Contour plots of static gauge pressure at the symmetry plane are displayed in Fig. 4b. In the beginning, the low pressure at the coalescence bridge draws the liquid to it which rapidly expands (Panel i). The span in the $x$-direction increases, followed by a recoil with a downward tendency due to the higher curvature at the $+y$ end than the bottom. The recoil is asymmetric and biased towards $-x$ as the higher wettability below Droplet D1 restricts liquid motion (Panel ii). This x-recoil in turn increases the span in the $y$- and $z$-directions. As the liquid body elongates in the $y$-direction, the adhesion of the hydrophilic wetted region below Droplet D1 creates a locally concave liquid-vapour interface (Panel iii). The pressure difference from the asymmetric curvature further draws the liquid towards $-x$, providing most of the tangential momentum generation. This cycle repeats, and the liquid body recoils in the other direction ($z$), biased towards $-x$, and experiences another curvature asymmetry (not shown in Fig. 4b). See Supplementary

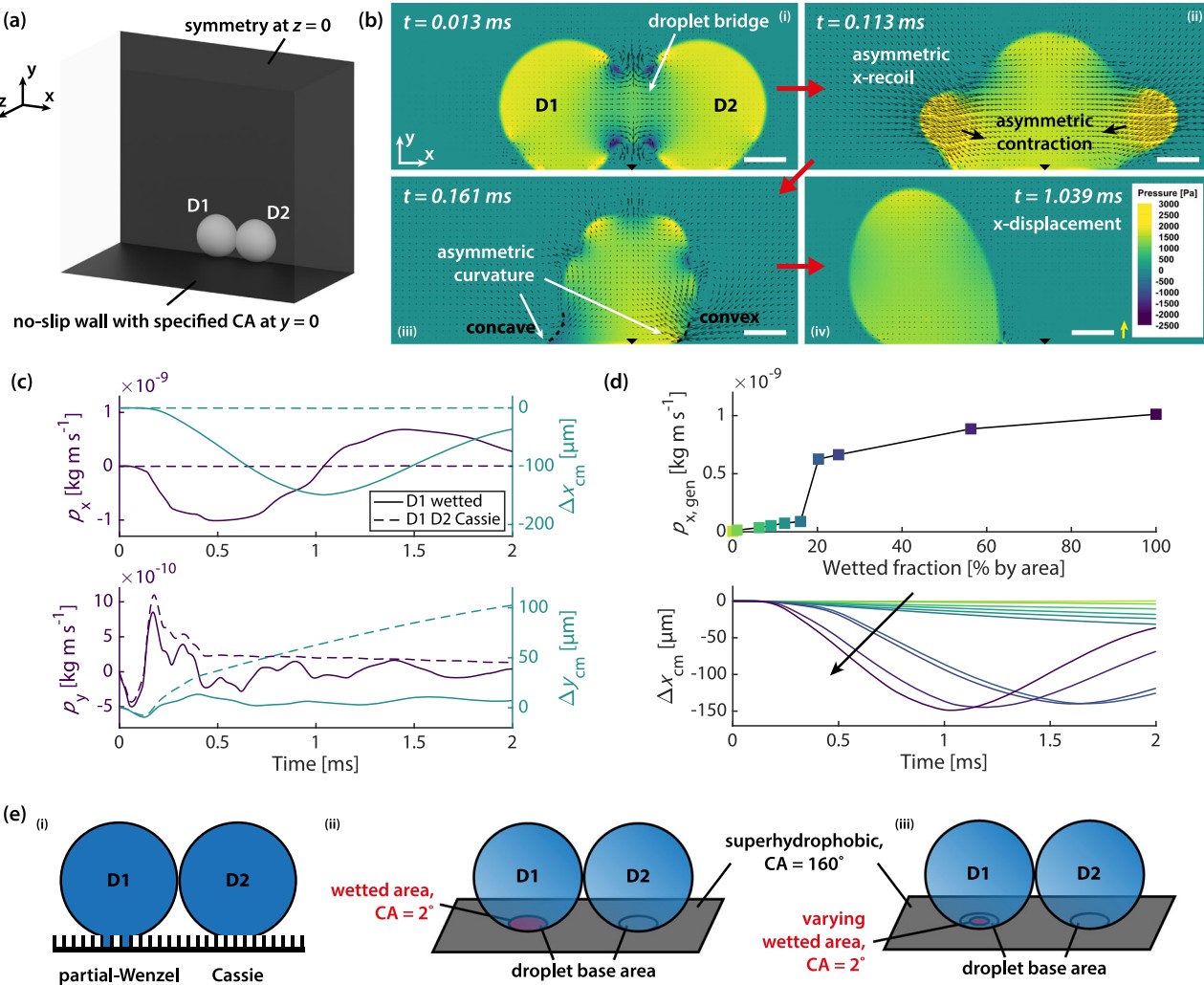

**Fig. 4 | Generation of tangential momentum. a** Computational domain. Two droplets with diameter 160 μm are placed on a no-slip wall at $y = 0$, specified with a contact angle. A symmetry plane is at $z = 0$. **b** Contour plots of static gauge pressure at the symmetry plane. The entire base area of Droplet D1 is wetted. Vectors are velocities. Scale bars: 50 μm. Yellow reference velocity vector: 2 m s$^{-1}$. **c** Momentum ($p_x$ and $p_y$ on the left $y$-axis) and centre-of-mass displacement ($\Delta x_{cm}$ and $\Delta y_{cm}$ on the right $y$-axis) in the $x$- and $y$-directions, for the case in which the base area of Droplet D1 is wetted and the case in which both Droplets D1 and D2 are in the Cassie state. **d** Maximum tangential momentum generated, $p_{x,gen} = \max(|p_x|)$, for varying

wetted fractions of the base area of Droplet D1 (top), and the corresponding $x$-centre-of-mass displacement (bottom). In **c** and **d**, the momentum reported reflects full spherical droplets, taking domain symmetry into account. **e** Numerical model and simulation cases. (i) Simultaneous presence of droplets at different wetting states. (ii) To mimic the effect of wetted nanostructures, the contact angle for the base area of D1 is set to 2°. (iii) The size of the wetted area of D1 is varied, and the remaining base area of D1 is kept at 160°, the same as the outer surface. Source data are provided as a Source Data file.

Information S13. Maximum $x$-displacement is reached at ≈1 ms (Panel iv). In the current case, we do not specify any dewetting step. The liquid body then swings back towards $+x$.

The complete event for a duration of 2 ms is shown in Supplementary Movie 6, together with the reference case when both Droplets D1 and D2 are in the Cassie state. The evolution of momentum and centre-of-mass displacement in the $x$- and $y$-directions is plotted in Fig. 4c against the reference case. In the latter, $x$-momentum and displacement remain zero whereas $y$-displacement increases continuously owing to droplet jumping. In the former, where the base of Droplet D1 is wetted, the absolute $x$-momentum reaches a maximum at ≈0.5 ms as the centre of mass of the system approaches the wetted region, ceasing tangential momentum generation. This mechanism reveals how wetting asymmetry due to increased adhesion of one droplet can serve as a hinge[24] to generate tangential momentum. Lastly, we vary the diameter and thus area of the wetted region as a percentage of the total base area below Droplet D1 and obtain the maximum tangential momentum generated for each percentage

(Fig. 4d). A sharp transition at ≈20% reveals that a slightly wetted base area can already generate substantial tangential momentum. See Fig. 4e for a summary of the numerical model and simulation cases.

## Dewetting and procession of roaming

After generating tangential momentum from asymmetric droplet adhesion, to be able to roam, the droplet has to dewet and detach from its location. In Fig. 5a, we dewet the surface by reverting the specified contact angle of the wetted region (2°) back to the original (160°) at 179 μs (Panel i), when the force exerted on it in the $+y$-direction is at maximum. After recoiling in the $z$-direction (Panel ii), the coalesced droplet departs with a substantial tangential component (Panel iii). The departure angle is sensitive to the dewetting time, as dewetting 20 μs later already results in entirely tangential departure. See Supplementary Movie 7 and Supplementary Information S14. The actual moment for dewetting depends on when both (1) static friction of the contact line[51] and (2) the adhesion work on the nanostructures are overcome.

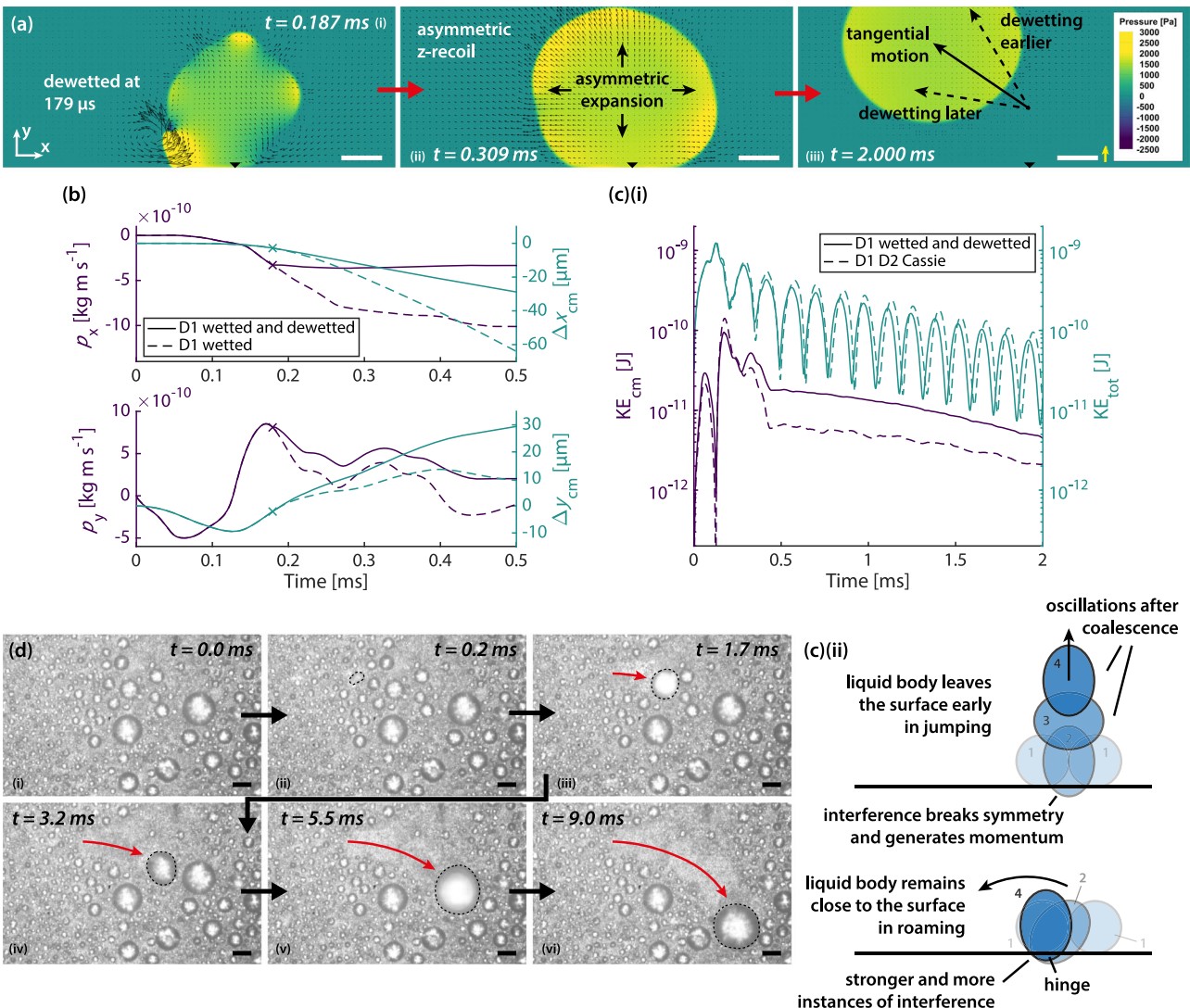

**Fig. 5 | Dewetting in roaming. a** Contour plots of static gauge pressure at the symmetry plane after dewetting at 179 μs. Initially the entire base area of Droplet D1 is wetted, similar to Fig. 4b. Vectors are velocities. If the droplet had dewetted at a different time, the x-component of the resultant motion would have been different as well (dashed arrows, also see Supplementary Fig. 31b). Scale bars: 50 μm. Yellow reference velocity vector: 2 m s⁻¹. **b** Momentum ($p_x$ and $p_y$ on the left y-axis) and centre-of-mass displacement ($\Delta x_{cm}$ and $\Delta y_{cm}$ on the right y-axis) in the x- and y-directions, for the case in which the original base area of Droplet D1 is subsequently dewetted at 179 μs, and the case in which it remains wetted. **c** (i) Kinetic energy of the translational motion of the centre of mass $KE_{cm}$ and the total kinetic energy $KE_{tot}$ for the two cases. The momentum and kinetic energy reported in **b** and **c** (i) reflect full spherical droplets, taking domain symmetry into account. **c** (ii) Schematic illustrating the symmetry breaking in jumping and roaming motions. When two droplets coalesce, the liquid body oscillates (numbered 1–4, 3 omitted in roaming for clarity). The interference with the surface breaks symmetry and generates momentum. In jumping, the liquid body leaves early and oscillates in the vapour. In roaming, the liquid body remains close to the surface. Oscillations interfere stronger with the surface repeatedly. The hinge then converts the symmetry breaking in the normal direction to a tangential direction. **d** Experimental observation of dewetting as roaming progresses. Coalescence is seen at 0.2 ms (Panel ii). Dewetting is seen at 1.7 ms (Panel iii) and 5.5 ms (Panel v) as indicated by the change in reflection of the main droplet. Black dashed lines enclose the main droplet. Red arrow indicates the approximate trajectory of the roaming event. Subcooling: 2.0 K. Gravity is downwards. Scale bars: 100 μm. Source data are provided as a Source Data file.

We plot the variation of momentum and displacement in Fig. 5b for the first 0.5 ms. After dewetting, tangential momentum generation ceases, but recovers in the normal direction. The kinetic energy of the translational motion of the centre of mass[50] compared to the total kinetic energy is expressed in Fig. 5c(i). Although the total kinetic energy of our dewetting and the reference cases are similar, the former exhibits a higher centre-of-mass translational kinetic energy, indicating a higher efficiency in producing centre-of-mass motion instead of oscillatory viscous dissipation. This applies to other dewetting times as well (Supplementary Fig. 32). The adhesion keeps the liquid body close to the surface, resulting in stronger and more instances of symmetry breaking than droplet jumping (Fig. 5c(ii)). The symmetry breaking is

converted into motion with a substantial tangential component using the adhesion asymmetry as a hinge, providing most of the in-plane momentum in roaming (Supplementary Information S14). In these simulations, we have selected the simplest case, the coalescence of binary droplets, such that the two cases (tangential and out-of-plane momentum generation) only differ in the presence of an adhesion asymmetry. We then showed how the generation of tangential momentum under such asymmetry can be more efficient than that of out-of-plane momentum observed in experiments, where more droplets are involved.

Finally, we demonstrate the dewetting experimentally in Fig. 5d. Coalescence can be seen at 0.2 ms. At 1.7 ms (Panel iii), the main

droplet is slightly lifted from the surface with a change in droplet reflection while maintaining significant tangential momentum. It is then intercepted by droplets on the surface and returns to the surface at 3.2 ms. In this event, we observe another dewetting at 5.5 ms (Panel v), before returning at 9.0 ms. See Supplementary Movie 8 for more examples. These changes in reflection appear when the droplet is slightly lifted from the surface during dewetting, much like the simulation shown in Fig. 5a. However, depending on the dewetting time, the droplet may adopt a more tangential motion (Supplementary Fig. 31b) and result in less prominent lifting and reflection changes. See Supplementary Information S14 for further discussion in interception.

In this work, we have focused on solely nanostructured superhydrophobic surfaces to eliminate the effects from microstructures. However, roaming on hierarchical surfaces might be possible (Supplementary Information S16). Roaming events in our work travel over distances below 1 mm, suggesting that roaming might also be possible on highly curved surfaces. The tangential momentum generation of coalescing condensate microdroplets on solely nanostructured superhydrophobic surfaces is attributed to the stochastic wetting state of the condensing droplets. The simultaneous presence of droplets at different wetting states results in adhesion asymmetry during coalescence, effectively converting excess surface energy to tangential kinetic energy as coalescence occurs. The ability of the coalesced droplet to dewet from the surface triggers roaming while preventing condensate flooding of the surface. This frequently renews the surface for fresh nucleation. Remarkably, this process significantly improves heat transfer compared to other condensate removal modes, as it takes over as the dominant mechanism with jumping dropwise condensation subsiding at higher subcooling.

## Methods

### Formation of boehmite nanowalls[27–29]
All aluminium substrates are of EN AW-1050A. The substrates are sonicated in acetone, isopropanol, deionised water for 10 min respectively, followed by sonication in 0.25 M sodium hydroxide solution for at least 10 min, before rinsing with deionised water and drying with nitrogen. The samples are then placed in hot water at ≈96 °C for 10 min and dried with nitrogen.

### pPFDA coating with iCVD[2,15]
The samples are first treated with oxygen plasma (Femto, Diener electronic) at 0.6 mbar for 10 min, followed by coating with trichlorovinylsilane (Sigma-Aldrich, CAS No.: 75-94-5) in a custom CVD chamber (saturated silane vapour at room temperature, ≈60 Torr). The samples are then placed in an iCVD system (iLab, GVD), to form a pPFDA coating at 100 mTorr using tert-butyl peroxide (Sigma-Aldrich, CAS No.: 110-05-4) as the initiator and 1H,1H,2H,2H-perfluorodecyl acrylate (Sigma-Aldrich, CAS No.: 27905-45-9) as the monomer. The stage and filament temperatures are set to 40 °C and 300 °C respectively. When applied on a pristine silicon wafer, the coating gives an advancing contact angle, contact angle hysteresis, and static contact angle of $124.2° ± 0.4°$, $12.5° ± 2.1°$ and $119.2° ± 1.5°$, respectively.

### Contact angle goniometry
Advancing and receding contact angles are measured with a goniometer (OCA 35, DataPhysics Instruments). Deionised water is deposited and withdrawn at a rate of $2\,\mu L\,s^{-1}$. The sample is blown dry with nitrogen before deposition of every droplet. Three measurements are taken before and after condensation. No significant change in wettability is observed. The static contact angle is computed from the mean of advancing and receding contact angles as a single droplet cannot be stably deposited.

### Numerical simulations
Cases are set up and computed with Ansys ICEM CFD and Ansys Fluent using the volume of fluid method. Saturation properties at 30 mbar are specified for the fluids. Postprocessing is performed in Tecplot 360 EX and MATLAB (MathWorks). See Supplementary Information S15 for details.

## Data availability
Experimental and simulation data are provided as a **Source Data** file with this paper and in the repository figshare (https://doi.org/10.6084/m9.figshare.28147487). Source data are provided with this paper.

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

## Acknowledgements

We thank Tobias Neef for his assistance with the iCVD process, and Jovo Vidic and Peter Feusi for their assistance in the construction of the condensation setup. We thank Henry Lambley and Jonathan Boreyko for helpful discussions; and Thibaut Delafosse and Mithulan Vasan for assisting in preliminary experiments. We thank Jiayu Song for preparing the titanium samples. We thank the Cleanroom Operations Team of the Binnig and Rohrer Nanotechnology Center (BRNC) for their help and support. Unless otherwise specified, fluid properties are obtained with CoolProp (www.coolprop.org)[52]. This project has received funding from the European Union's Horizon 2020 research and innovation programme under grant number 801229 (HARMoNIC). C.W.E.L. acknowledges funding from the Croucher Foundation during revision of this manuscript.

## Author contributions

C.W.E.L. and D.P. conceived the research. D.P. supervised all aspects of the research and provided scientific guidance. C.W.E.L. designed and constructed the condensation setup, conducted the experiments, performed the simulations, and analysed the data. C.W.E.L. prepared the aluminium samples. M.D. prepared the copper samples. C.W.E.L., K.R., and A.T. applied the pPFDA coatings. G.C.P. assisted in the simulations. D.P., K.R., C.S.S., and A.M. provided scientific guidance for the various aspects of the research. C.W.E.L. and D.P. wrote the manuscript with contribution from all other authors.

## Funding

## Competing interests

The authors declare no competing interests.
