## [Transparent Peer Review file · Nature Communications]

Condensate droplet roaming on nanostructured superhydrophobic surfaces

Corresponding Author: Professor Dimos Poulikakos

Version 0:

Reviewer comments:

Reviewer #1

(Remarks to the Author)

In this research, the authors focused on the roaming phenomenon of coalescing condensate microdroplets on superhydrophobic surfaces and the underlying mechanism is presented. They further show that the roaming process significantly improved heat transfer efficiency compared to condensate droplet jumping. This manuscript is well-rewritten and reads well. I support its publication in Nature Communications.

Reviewer #2

(Remarks to the Author)

I agree with the authors that the physics of roaming on these nanostructured surfaces seem to be distinctly different from previously described lateral movement during condensation, i.e., that this finding justifies publication in a reputable journal such as NatComm (even NatPhys, in my opinion). The hypothesis and conclusions also seem reasonable. The most plausible alternative explanation to the observations would be a motion triggered by temperature gradients (which might also very well come from partial nucleation!), similar to that observed by Zhu et al (Appl Phys Lett 116, 243703 (2020)). All my other previous comments have been addressed and I can recommend this manuscript for publication

Reviewer #3

(Remarks to the Author)

The current study investigates the mechanism regarding an observed in-plane droplet movement during coalescence events on a superhydrophobic surface unique to using solely nanostructures. The authors show that their “roaming” condensation provides a higher condensation heat transfer performance than jumping droplet condensation as roaming droplets can clear more area at a faster rate than jumping events. Providing droplet removal aids in making more area available for renucleation to occur thus enhancing condensation.

Comments and questions:

1. The major novelty highlighted in the current study concerns the investigation of roaming droplet condensation and its effects on condensation heat transfer performance. While the phenomenon is intriguing, it does not appear to be entirely new. Rather, the enhancement of condensation heat transfer seems to be the key innovation here. The authors should emphasize this aspect to avoid disputes regarding the so-called “roaming” droplet condensation novelty. In reference [23], the sweeping phenomenon mentioned seems like what is termed roaming in the current study. Reference [23] utilized hierarchical structures and focused more on jumping droplet condensation, but it appears that the current study identified a higher-performing condensation mode using only nanostructures.

However, a major concern with this study is the heat transfer data presented in Figure 3. A common issue with superhydrophobic surfaces is their tendency to flood at higher subcooling due to nucleation occurring in their underlying micro and nanostructures (10.1021/nl303835d). Specifically, in Figure 3, the superhydrophobic surface data is only plotted up to a subcooling of around 3 K. The authors describe this as “high” subcooling, yet they plot filmwise condensation up to 10 K. Was flooding observed on the superhydrophobic surfaces at higher subcoolings? Additionally, the heat transfer coefficient trend does not align with typical experimental results in the literature. For example, at low subcooling, the heat

transfer coefficient is usually the highest, and as subcooling increases, the heat transfer coefficient decreases (10.1002/adfm.202211113). This is typically because condensation rates exceed droplet removal rates at higher subcoolings. Can the authors explain why their heat transfer coefficient only increases?

2. It appears that no experimental validation was conducted for the heat transfer measurements. Typically, a comparison is made between the theoretical Nusselt filmwise condensation model and experimental filmwise condensation data. An example can be found in Figure 7 (10.1016/j.ijheatmasstransfer.2014.11.069). Such experimental validation helps to ensure the reliability of heat transfer measurements.

3. Is the main mode of droplet removal still through gravity-driven shedding? It does not appear this was discussed.

4. In Figure 3, the superhydrophobic surface is labeled as exhibiting jumping droplet condensation and roaming condensation as subcooling increases. Can the authors explain why the mechanism would start as jumping droplet condensation and then transition to roaming condensation? This transition is not quite clear. In Movie S5, it appears at low subcooling that the “jumping” resembles a faster version of roaming condensation. Can the authors clarify this observation?

5. It is unclear what additional information the simulation results in Figures 4 and 5 provide that is not already shown in the experimental videos. Droplet coalescence events and the displacement of droplets can be easily observed in the videos. It is also confusing why a no-slip boundary condition was used in the simulation, but then a 2° contact angle at the base of the droplet is applied. How does this contact angle interact with the base if there is a no-slip boundary condition? Furthermore, the simulation clearly shows droplet jumping in Figure 5, so why is this referred to as roaming? At least, roaming is not a controllable phenomenon as jumping is inevitable. Recently, a coarsening droplet is reported to significantly enhance condensation in a controlled way (10.1002/dro2.23)

6. In the supporting movies S3 and S4, the phenomena described in the titles are not directly shown. In Movie S3, “Roaming events altering direction,” the roaming droplet simply moves to the left and never changes direction. In Movie S4, “Roaming events terminating in jumping,” the video never shows a droplet jumping at the end of roaming. If it does, it is not clear. These videos are misleading as they do not show the supposed behavior described in the manuscript.

7. Regarding the heat transfer comparison made in Figure S8.2, it is great that PDMS brushes were compared to the current work, as these are state-of-the-art. Ideally, it would be best to compare results taken from the same experimental set-up, but it is understandable that this is challenging. However, it may not be a fair comparison based solely on subcooling. It may be good to compare the heat transfer coefficient versus heat flux as well.

8. In the current study, subcooling is defined as $\Delta T = T_{\text{steam}} - T_{\text{surf}}$. The term T_{steam} should be replaced with T_{sat} , where the value of subcooling is directly defined based on the saturation temperature of steam, depending on the thermodynamic state of the chamber conditions.

9. The supporting videos are too large. There are ways to reduce the size of each video significantly while maintaining the resolution.

Reviewer #4

(Remarks to the Author)

In this work, the authors report the in-plane roaming of condensate droplets upon coalescence on nanostructured superhydrophobic surfaces. Despite the authors' claims, the novelty and significance of this work are questionable. The authors need to address the below comments before the manuscript can be accepted for publication.

1. The authors themselves mention and cite references that tangential motion of condensate droplets has been well-studied previously on microstructured surfaces. Here, the authors report a similar phenomenon on nanostructured surfaces, terming it “roaming droplets.” Thus, the work has limited novelty and uniqueness.

2. Despite the detailed discussion, it is not clear why in-plane roaming condensation cannot occur at a lower degree of subcooling. Even at lower subcooling, the probability of droplets jumping upon coalescence is low, and the asymmetry during coalescence still exists. Why is there no roaming effect at lower subcooling?

3. The discussion on the limitations of roaming condensation is significantly lacking. Jumping droplet condensation can occur on a wide range of surfaces (nanostructured, hierarchical, etc.). However, roaming condensation seems to be possible solely on nanostructured surfaces with unique surface morphologies.

4. The current work focuses on condensation on superhydrophobic surfaces. However, the introduction discusses lubricant-infused surfaces at length and cites several references, which is unwarranted.

5. The authors emphasize the higher heat transfer coefficient (HTC) for roaming condensation versus jumping droplet condensation. This is not a fair comparison. Generally, the heat transfer coefficient increases with increasing subcooling during dropwise condensation. Roaming condensation is realized only at higher subcooling, while jumping droplet condensation occurs at lower subcooling. How do the authors decouple the effects of subcooling degree and condensation mode on HTC?

6. In the “generation of tangential momentum in roaming” section, the authors state “without this, there is no apparent reason for out-of-plane jumping to in-plane roaming.” This statement is speculative and lacks sufficient explanation and support.

7. In-plane roaming condensation should induce a substantial sweeping effect, clearing out condensate droplets leading to rapid regeneration of nucleation sites. However, there is hardly any discussion of this effect.

8. Condensation on superhydrophobic surfaces also induce hierarchical condensation, where smaller nucleating droplets within the surface structures are absorbed by the larger sink droplet on the surface. Can such events induce the roaming effect?

9. How would the roaming condensation dynamics change if the condensing surface were a tube instead of a flat surface? Additionally, if the surface morphology changes from nanostructured alone to hierarchical structures (e.g., microstructured bumps with nanoneedles), how would roaming condensation be affected?

Version 1:

Reviewer comments:

Reviewer #3

(Remarks to the Author)

The authors have addressed all my comments. I recommend it for publication and congratulate the authors for their publication.

Reviewer #4

(Remarks to the Author)

The authors have addressed most of the queries raised, significantly enhancing the quality of the manuscript. However, a few additional clarifications would be beneficial.

1) It is understood that, on a given surface, decoupling the relationships between heat transfer coefficients and subcooling, as well as between condensation behavior (mode) and subcooling, is challenging. The initial comment stemmed from this complexity. Specifically, the authors compare heat transfer coefficients in the jumping droplet regime to those in the roaming droplet regime, suggesting enhancement. This comparison warrants caution, as multiple factors could influence the results, and the enhancement may not solely be due to roaming droplets. Including a sentence or two in the discussion to address this nuance would be valuable.

2) Hierarchical condensation is observed across a broad range of droplet sizes (Yan et al., ACS nano 13.7 (2019): 8169-8184.), which includes those reported by the authors. Could the authors elaborate on why this phenomenon would not induce roaming droplets, beyond describing it as increasingly difficult and improbable?

Version 2:

Reviewer comments:

Reviewer #4

(Remarks to the Author)

The authors have addressed all my comments. I recommend it for publication.

Paper title: Condensate droplet roaming on nanostructured superhydrophobic surfaces

The original remarks by the reviewers are shown verbatim in blue in this document. Direct quotes from the manuscript and supplementary information are in green in this document. Major modifications made to the manuscript and/ or the supplementary information are highlighted both in this document and in the revised documents.

REVIEWER #1:

In this research, the authors focused on the roaming phenomenon of coalescing condensate microdroplets on superhydrophobic surfaces and the underlying mechanism is presented. They further show that the roaming process significantly improved heat transfer efficiency compared to condensate droplet jumping. This manuscript is well-rewritten and reads well. I support its publication in Nature Communications.

We thank the reviewer for the time in reviewing the manuscript and the positive opinion.

REVIEWER #2:

I agree with the authors that the physics of roaming on these nanostructured surfaces seem to be distinctly different from previously described lateral movement during condensation, i.e., that this finding justifies publication in a reputable journal such as NatComm (even NatPhys, in my opinion). The hypothesis and conclusions also seem reasonable. The most plausible alternative explanation to the observations would be a motion triggered by temperature gradients (which might also very well come from partial nucleation!), similar to that observed by Zhu et al (Appl Phys Lett 116, 243703 (2020)).

All my other previous comments have been addressed and I can recommend this manuscript for publication

We thank the reviewer for the time in reviewing the manuscript and the positive opinion. The work of Zhu et al.¹ points to an interesting pathway for spontaneous condensate microdroplet motion. However, we believe that the mechanism presented is different from roaming.

In the paper,¹ a non-uniform temperature field arises by condensation on a heterogeneous lubricant-infused surface. This non-uniform field powers droplet motion through a thermocapillary (Marangoni) effect. As such, an individual droplet can move spontaneously across the surface, coalescing with smaller droplets on its path. Motion of millimetric droplets up to 4.3 mm¹ is observed. The observed velocity is relatively low ($\sim 0.08 \text{ mm s}^{-1}$ for a droplet of 1.7 mm in diameter).¹

In contrast, roaming occurs exclusively on microdroplets as it is fundamentally powered by coalescence and the corresponding release of excess surface energy. Roaming events only initiate upon contact and coalescence of two or more droplets. The time scale of energy conversion for a typical event is significantly shorter: The mean duration of a roaming event is 5.3 ms, compared to the motion spanning on the order of seconds described in their work.¹ This rapid conversion of energy results in a significantly higher (4 orders of magnitude) droplet velocities at $\approx 0.18 \text{ m s}^{-1}$ as described in the manuscript. Lastly, the thermocapillary motion in their work¹ occurs on a lubricant-infused slippery surface. There is significant contact between the droplet and the surface (due to lower contact angles) and temperature nonuniformity of the surface could influence surface tension nonuniformity of the droplet and lubricant layer, leading to motion from Marangoni flows. Roaming, on the other hand, occurs on superhydrophobic nanostructured surfaces which have minimal contact with the droplets. Therefore, temperature nonuniformity, if any, would influence the physics mostly through the intricate interaction with the nucleation dynamics.

We have revised the manuscript (p. 3) to include the work of Zhu et al.¹ as a pathway to improve dropwise condensation performance:

The pursuit for efficient heat removal has strongly motivated surface engineering research, with the central idea to minimise the residence time and amount of the liquid condensate on the cooled surface.¹⁻

10

REVIEWER #3:

The current study investigates the mechanism regarding an observed in-plane droplet movement during coalescence events on a superhydrophobic surface unique to using solely nanostructures. The authors show that their “roaming” condensation provides a higher condensation heat transfer performance than jumping droplet condensation as roaming droplets can clear more area at a faster rate than jumping events. Providing droplet removal aids in making more area available for renucleation to occur thus enhancing condensation.

We thank the reviewer for the time and effort in reviewing our manuscript. We hope the following responses can provide satisfactory clarification to our work for its reconsideration.

Comments and questions:

1. The major novelty highlighted in the current study concerns the investigation of roaming droplet condensation and its effects on condensation heat transfer performance. While the phenomenon is intriguing, it does not appear to be entirely new. Rather, the enhancement of condensation heat transfer seems to be the key innovation here. The authors should emphasize this aspect to avoid disputes regarding the so-called “roaming” droplet condensation novelty. In reference [23], the sweeping phenomenon mentioned seems like what is termed roaming in the current study. Reference [23] utilized hierarchical structures and focused more on jumping droplet condensation, but it appears that the current study identified a higher-performing condensation mode using only nanostructures.

The observation of spontaneous tangential droplet motion of condensate droplets upon coalescence on a nanostructured superhydrophobic surface has previously been reported in literature.^{11,12} We have cited all relevant studies to our knowledge in the manuscript. However, these studies have not identified and explained the unique physics underlying such motion. For example, Ref.¹² (Reference [23] mentioned by the reviewer) has attributed the emergence of tangential motion to the lack of sufficient kinetic energy released from coalescence of partially wetted droplets for them to jump. We quote verbatim the argument from Ref.¹² as follows:

“The kinetic energy of the sweeping droplet comes from the free energy released during droplet coalescence, which is insufficient to directly push the droplet off the surface by self-propelled jumping.” (p. 44915)

“Limited by the insufficient kinetic energy released during the coalescence of droplets in the partially wetting state, the merged droplets cannot jump off but rapidly slip on the condensing surface.” (p. 44918)

It is true that the adhesion of partially wetted droplets is higher relative to the Cassie state. However, this in itself does not readily transform normal jumping motion into tangential roaming motion. This is due to two main reasons:

- (1) There is no known mechanism to convert excess surface energy into kinetic energy in the tangential direction. In the above argument, Ref.¹² has suggested that the excess energy has to “push the droplet off” for it to jump. In the tangential direction, there is no surface to push against.
- (2) Both jumping and roaming require the partially wetted droplets to dewet so that they can depart from their starting location. If the excess surface energy of coalescence has indeed become unable to overcome the increased adhesion, then the droplet cannot depart normally by jumping; but it cannot depart tangentially by roaming either.

We have modified the presentation in the manuscript to bring about the two points above (p. 17):

The normal impingement of the bridge can only generate a normal reaction from the surface in the opposite direction. For roaming, there is no symmetry-breaking surface for a tangential reaction force to

manifest as the condensate microdroplets are orders of magnitude larger than the underlying structures. If the increase in adhesion for all condensate droplets on the surface were uniform when subcooling increases, i.e. no asymmetry, jumping would gradually cease and **transition to flooded condensation would be directly evident without any intermediate in-plane roaming regime**. The excess surface energy from coalescence would no longer overcome the increased adhesion and be dissipated instead. Droplets would not depart at all, whether in-plane or out-of-plane, until they attain the size when gravity dominates.

In our work, we have provided a pathway for roaming motion to occur from excess surface energy of coalescence. We explain that the asymmetry in adhesion brought about by stochastic nucleation creates a hinging effect in which the momentum generated can be directed to the tangential direction. There is an increase in adhesion, but the increase is not uniform over all droplets. The excess surface energy of coalescence could still overcome the adhesion, but being non-uniform, the released energy generates substantial tangential momentum instead. We then show that the lively roaming mechanism is more efficient and enhances heat transfer efficiency. Our work is the first in at least the following facts:

1. Identifying roaming motion of coalescing condensate microdroplets on solely nanostructured superhydrophobic surfaces, which indicates a different tangential momentum generation mechanism from the ones previously understood;
2. Observing evolving roaming motion with sufficient temporal and spatial resolution for characterisation from initiation to termination;
3. Quantitatively characterising roaming including propagation velocities and event lengths;
4. Simultaneous in-situ measurement of heat transfer, showing that roaming condensation provides the highest heat transfer coefficient, even higher than jumping dropwise condensation; and
5. Providing a mechanism for tangential momentum generation arising from the spontaneous wettability asymmetry when condensation occurs stochastically within nanostructure cavities. Currently available studies have only pointed to increased adhesion, which is itself insufficient for producing tangential motion.

Therefore, our work is not only novel in showing heat transfer enhancement, as the reviewer suggested, but also in the physics of nucleation and droplet coalescence dynamics.

However, a major concern with this study is the heat transfer data presented in Figure 3. A common issue with superhydrophobic surfaces is their tendency to flood at higher subcooling due to nucleation occurring in their underlying micro and nanostructures (10.1021/nl303835d). Specifically, in Figure 3, the superhydrophobic surface data is only plotted up to a subcooling of around 3 K. The authors describe this as “high” subcooling, yet they plot filmwise condensation up to 10 K. Was flooding observed on the superhydrophobic surfaces at higher subcoolings?

In the accessible range of subcooling for the superhydrophobic boehmite in our work, we do not observe any signs of flooding. However, we agree with the reviewer that superhydrophobic surfaces tend to flood at higher subcooling levels. In fact, all superhydrophobic surfaces flood at a sufficiently high subcooling. This is because nucleation within the structure cavities is inevitable, as the nucleation diameter decreases and eventually becomes smaller than the structure cavity sizes when subcooling is increased. For instance, for the steam conditions in this work (saturated steam at 30 mbar), the critical nucleation diameter at a subcooling of 10 K is only 3.5 nm (**Figure R3.1**), which is an order of magnitude smaller than the size of nanostructure cavities of boehmite.

Figure R3.1: Variation of critical nucleation diameter d_{crit} with subcooling up to 10 K, for condensation from saturated steam at 30 mbar. Calculated from Ref.¹³.

While it is possible to engineer a superhydrophobic surface to withstand such low nucleation diameters without flooding, it is not the focus of our work as we seek to understand the mechanism of roaming condensation and how to exploit its heat transfer benefits.

First, there is no objective standard for a “high” subcooling, given that the control always exhibits filmwise condensation for any subcooling. In other words, the control can be tested at arbitrarily high subcooling levels, and for any superhydrophobic surface, there will always be a threshold at which it begins to flood. In this work, we aim to show that on superhydrophobic surfaces, roaming condensation occurs at higher subcooling than jumping dropwise condensation, with even higher heat transfer coefficients. We claim that superhydrophobic surfaces can be operated in this regime, to take advantage of this extended subcooling window together with its higher heat transfer efficiency, before flooding sets in. In **Figure S10.4** (p. 51 of Supplementary Information, reproduced below), we have presented how condensation can be targeted for optimal heat transfer performance on surfaces of different structure characteristics and flooding resistance:

Figure S10.4: Control and promotion of roaming. Sparser nanostructures lower the transition subcooling and roaming occurs at lower subcooling levels. Enhanced flooding resistance on the surface structure level extends the subcooling range to higher levels. Heat transfer efficiency is maximised when roaming occurs at high subcooling levels, which should be targeted, as shown for example in the top bar of the figure.

The specific choice of a particular superhydrophobic surface in our work is not a matter of concern for the phenomenon is general. In this work, we have focused on superhydrophobic boehmite for several reasons. (1) It is based on aluminium, which bears relevance to heat transfer applications; (2) It has relatively well-defined structures composed of closed cavities with measurable sizes; and (3) It is reflective enough for high-speed imaging. For completeness, we have presented the jumping dropwise, roaming, and flooded condensation regimes in **Figures S10.2** and **S10.3** (p. 49 and 50 of Supplementary Information, reproduced below) on superhydrophobic copper(II) hydroxide nanoneedles. Their sparser nanostructures result in flooding at lower subcooling, and it is observable in our accessible subcooling range for this surface.

Figure S10.2: Heat transfer at different subcooling on superhydrophobic copper(II) hydroxide nanoneedles. (a) Measured heat fluxes q'' at steady state. Lines of constant heat transfer coefficient h are shown in grey, from 20 to 90 kW m⁻² K⁻¹ at intervals of 10. (b) Heat transfer coefficients h corresponding to a. Lines of constant heat flux q'' are shown in grey, from 25 to 175 kW m⁻² at intervals of 50.

Figure S10.3: Snapshots of condensation behaviour at different subcooling on superhydrophobic copper(II) hydroxide nanoneedles. Jumping dropwise condensation is seen in **a**, where the droplets which have jumped away from the surface block the droplets on the surface from view. Transition is seen from (a) jumping dropwise to (b) roaming condensation, and finally to (c) and (d) flooded condensation. Scale bars: 500 μm .

The superhydrophobic boehmite will eventually exhibit flooding given sufficiently high subcooling, as all superhydrophobic surfaces would, and we expect a jumping dropwise-roaming-flooding transition. In **Figure 3a**, the flooding regime for superhydrophobic boehmite does not lie in its accessible range of subcooling (< 2.2 K). We would like to point out that the accessible range is dependent on the tested surface. When a surface is tested, its temperature (thus the subcooling) cannot be directly imposed. We vary a thermal boundary condition of the system instead and measure the resultant surface temperature and subcooling. This concept can be visualised with the schematic in **Figure R3.2**. Our experimental setup is shown at the top, with the temperature gradient in the middle the thermal resistance circuit at the bottom. The boundary conditions are the cooler back end temperature $T_{\text{coolerback}}$ and the steam temperature T_{steam} . Thermal resistances come from the cooler copper cylinder, thermal paste, sample thickness, and the condensation.

Figure R3.2: Experimental setup (top), temperature gradient (middle), and the thermal resistance circuit (bottom).

In **Figure 3a**, we did not deliberately choose to plot filmwise condensation up to a subcooling of 10 K: It is a direct consequence of a sample surface with lower heat transfer coefficients. In **Figure R3.2**, for given cooler back end and steam temperatures, the surface temperature T_{surf} (and the subcooling) is determined by the relative magnitude of thermal resistances composing the circuit. When we test different samples, the condensation thermal resistance ($= 1/h$) is different. Therefore, the condensation efficiency of the sample surface directly determines the subcooling achieved.

The steam temperature is fixed (saturation temperature at 30 mbar, 24.1 °C). To reach different subcooling levels, we impose the cooler back end temperature at 7 points from 20 to -10 °C (**Supplementary Information S2**, p. 15). For the lowest set cooler back end temperature (-10 °C), the subcooling reaches 2.2 K on the superhydrophobic boehmite and 10.1 K on the pristine boehmite. This is a direct result of the higher heat transfer coefficient of the superhydrophobic boehmite: Because it has a lower condensation heat transfer resistance ($= 1/h$), the surface temperature is closer to the steam temperature, resulting in a lower subcooling. In **Figure R3.3**, we plot the subcooling and heat transfer coefficient of the two surfaces against the cooler back end temperature.

Figure R3.3: Subcooling and heat transfer coefficients at the 7 cooler back end temperatures.

Lastly, although all superhydrophobic surfaces can only operate within a limited subcooling window, their superior efficiency often translates to more heat transferred than other surfaces operating at a substantially higher subcooling. As a matter of fact, our superhydrophobic boehmite achieves a heat flux of $\approx 160 \text{ kW m}^{-2}$ at a subcooling of 1.7 K, whereas the pristine boehmite would require 10.1 K (**Figure S8.1**, p. 37 of Supplementary Information, reproduced below). In practice, this suggests that it is less important to achieve “high” subcooling on efficient superhydrophobic surfaces.

Figure S8.1: Measured heat fluxes q'' at steady state. Lines of constant heat transfer coefficient h are shown in grey, from 10 to 110 $\text{kW m}^{-2} \text{K}^{-1}$ at intervals of 10. $dh/d(\Delta T) > 0$ when $dq''/d(\Delta T) > h$. On the superhydrophobic surface, two modes of condensation are observed.

We have added the above discussion to **Supplementary Information S2** (p. 21):

Accessible range of subcooling

The accessible range of subcooling is dependent on the tested surface. When a surface is tested, its temperature (thus the subcooling) cannot be directly imposed. We vary a thermal boundary condition of the system instead and measure the resultant surface temperature and subcooling. This concept can be visualised with the schematic in **Figure S2.4**. Our experimental setup is shown at the top, with the temperature gradient in the middle and the thermal resistance circuit at the bottom. The boundary conditions are the cooler back end temperature $T_{\text{coolerback}}$ and the steam temperature T_{steam} . Thermal resistances come from the cooler copper cylinder, thermal paste, sample thickness, and the condensation.

Figure S2.4: Experimental setup (top), temperature gradient (middle), and the thermal resistance circuit (bottom).

In **Figure 3a**, we did not deliberately choose to plot filmwise condensation up to a subcooling of 10 K: It is a direct consequence of a sample surface with lower heat transfer coefficients. In **Figure S2.4**, for given cooler back end and steam temperatures, the surface temperature T_{surf} (and the subcooling) is determined by the relative magnitude of thermal resistances composing the circuit. When we test different samples, the condensation thermal resistance ($= 1/h$) is different. Therefore, the condensation efficiency of the sample surface directly determines the subcooling achieved.

The steam temperature is fixed (saturation temperature at 30 mbar, 24.1 °C). To reach different subcooling levels, we impose the cooler back end temperature at 7 points from 20 to -10 °C (see above). For the lowest set cooler back end temperature (-10 °C), the subcooling reaches 2.2 K on the superhydrophobic boehmite and 10.1 K on the pristine boehmite. This is a direct result of the higher heat transfer coefficient of the superhydrophobic boehmite: Because it has a lower condensation heat transfer resistance ($= 1/h$), the surface temperature is closer to the steam temperature, resulting in a lower subcooling. In **Figure S2.5**, we plot the subcooling and heat transfer coefficient of the two surfaces against the cooler back end temperature.

Figure S2.5: Subcooling and heat transfer coefficients at the 7 cooler back end temperatures.

We have included the work of Miljkovic et al. in the manuscript (p. 3) to highlight the surface engineering required to minimise condensate accumulation and flooding for condensation heat transfer enhancement:

The pursuit for efficient heat removal has strongly motivated surface engineering research, with the central idea to minimise the residence time and amount of the liquid condensate on the cooled surface. ¹

10

Additionally, the heat transfer coefficient trend does not align with typical experimental results in the literature. For example, at low subcooling, the heat transfer coefficient is usually the highest, and as subcooling increases, the heat transfer coefficient decreases (10.1002/adfm.202211113). This is typically because condensation rates exceed droplet removal rates at higher subcoolings. Can the authors explain why their heat transfer coefficient only increases?

For a given surface at a given subcooling, there is a corresponding condensation behaviour, and its heat transfer efficiency is measured by the heat transfer coefficient. When the subcooling changes, the condensation behaviour on different surfaces responds and changes differently, resulting in the different trends of heat transfer coefficients with subcooling.

At steady state, regardless of condensation mode, the mean condensation rate always equals the mean condensate removal rate. Otherwise, condensate would indefinitely accumulate. When subcooling increases, the condensation rate initially increases. Then, there is a response in the removal rate, which is dependent on the surface. When the two rates balance out each other, condensation attains the steady state at this subcooling. The mean condensate distribution then determines the mean thermal resistance, and the steady state mean heat transfer efficiency and coefficient at this subcooling can be obtained.

For filmwise condensation, the heat transfer coefficient usually decreases when subcooling increases because the condensate film becomes thicker and imposes a larger thermal resistance. Our measurements on pristine boehmite in **Figure 3a** follow this expected trend. However, for condensation modes in which the condensate manifests as discrete droplets, there may not be a general trend for heat transfer coefficient against an increase in subcooling. The heat transfer coefficient for these surfaces mostly depends on (1) the nucleation rate and (2) the surface area renewal rate. This is because most heat transfers through the smallest droplets¹⁴ (a parallel thermal resistance circuit). At higher subcooling, there are higher nucleation rates and smaller nucleation diameters. Both are beneficial to heat transfer because there are more droplets for the vapour to directly condense on, and because of their smaller sizes there is less thermal resistance as well. Therefore, whether the heat transfer coefficient increases with subcooling depends on whether the higher nucleation rates and smaller nucleation diameters can be sustained. Below the transition subcooling of 1.5 K, the condensation mode on superhydrophobic boehmite is jumping dropwise condensation. Even when subcooling has increased from 0.7 to 1.3 K, it can be seen (second panel, **Figure 3b**, p. 15 of the manuscript, reproduced below) that most of the surface is still void of condensate droplets,

as the droplet removal mechanism is still efficient. The higher nucleation rate of smaller nuclei at this higher subcooling can occur in these empty spaces, enabling an overall lower thermal resistance and a higher heat transfer coefficient.

Figure 3: (b) Snapshots of condensation behaviour for superhydrophobic boehmite. Transition is seen from jumping dropwise to roaming condensation. At the lowest subcooling (0.7 K), only jumping is observed and there are numerous droplets in the vapour, with darker appearance and out-of-focus contour. These droplets in the vapour travel in one general direction to the bottom left due to steam flow (leftward) and gravity (downward). At 1.3 K, the number of jumped droplets in the vapour is visibly reduced, and some are seen to return to the surface. After the transition subcooling (1.5 K), condensation is dominated by roaming. Red arrows are trajectories of roaming events. Roaming droplets travel in all in-plane directions. See **Video S5** for the corresponding video. Scale bars: 500 μm .

After the transition subcooling, roaming condensation becomes the dominant condensation mode. In this regime, the heat transfer coefficient further increases with subcooling. There are more larger droplets on the surface (third and fourth panels, **Figure 3b**, p. 15 of the manuscript, reproduced above) than jumping dropwise condensation at subcooling < 1.5 K. However, roaming condensation renews the surface so rapidly that the even higher nucleation rate of even smaller nuclei can be sustained at these even higher subcooling levels. We have quantified the surface area renewed by roaming motion, the surface area renewal rate S' , in **Figure 3c**. Over 70% of the surface area is renewed every second, enabling ample space for fresh nucleation. Our results suggest that the droplet removal rates during jumping dropwise condensation and roaming condensation are sufficient to support an increase in heat transfer coefficients when subcooling increases, thus the trend for superhydrophobic boehmite in **Figure 3a**. We expect the trend of increasing heat transfer coefficient to continue with increasing subcooling, until the nucleation rate and the critical nucleation diameter pass the threshold and flooding of the structures ensues. As the structures flood, the removal mechanism fails, and condensation settles at a steady state with higher thermal resistance. The heat transfer coefficient gradually drops as the structures flood more, as seen in our measurements on superhydrophobic copper(II) hydroxide nanoneedles in **Figure S10.2b**.

From our experience, we have indeed observed heat transfer coefficients weakly decreasing with increasing subcooling for a (gravitational) dropwise condensing polytetrafluoroethylene before (PTFE in Figure 3a in Ref.¹⁵, same experimental setup as current work). However, for surfaces with better droplet removal abilities, we have observed increasing heat transfer coefficients with increasing subcooling for at least a range of subcooling. For example, on PDMS brushes (Figure 3b in Ref.¹⁶, same experimental setup as current work) and lubricant-infused surfaces (VG LIS in Figure 3a in Ref.⁵, same experimental setup as current work). Therefore, for condensation modes other than filmwise condensation (i.e. there are discrete droplets), the trend of heat transfer coefficient with subcooling is highly dependent on the surface type, mostly on its ability to remove droplets efficiently.

We have modified the manuscript (p. 13) for a more precise presentation, and included the work of Boylan et al.¹⁷ to highlight the possibility of different trends of heat transfer coefficients with subcooling:

The heat transfer coefficient can exhibit different trends with subcooling for surfaces on which condensate appears in the form of droplets.^{5,15-17} When droplet removal is sufficiently efficient, it can increase with subcooling,^{5,16} due to higher nucleation rates and smaller nucleation diameters. This

important trend is also present in the superhydrophobic boehmite heat transfer coefficients in **Figure 3a**. However, this trend sustains so long as the condensation mode allows it, that is, **there is limited flooding of structures or saturation of nucleation sites because of the efficient droplet removal. The > 70% area renewed every second in roaming condensation enables ample space for fresh nucleation, sustaining the trend.**

2. It appears that no experimental validation was conducted for the heat transfer measurements. Typically, a comparison is made between the theoretical Nusselt filmwise condensation model and experimental filmwise condensation data. An example can be found in Figure 7 (10.1016/j.ijheatmasstransfer.2014.11.069). Such experimental validation helps to ensure the reliability of heat transfer measurements.

The heat transfer coefficients at different subcooling for filmwise condensation can be compared to the Nusselt model for validation. For condensation from saturated steam on a small vertical plate, it can be calculated as (Equation 9-18 in Ref.¹⁸):

$$h = 1.13 \left[\frac{\rho(\rho - \rho_{\text{steam}})gh_{fg}k^3}{L\mu(T_{\text{steam}} - T_{\text{surf}})} \right]^{\frac{1}{4}}$$

where h_{fg} is the latent heat of vaporisation at 30 mbar (= 2443.9 kJ kg⁻¹). L is the length of the vertical condensing area in the gravitational direction (= 20 mm). ρ_{steam} is the density of the steam (= 0.021904 kg m⁻³). ρ , k and μ are the density, thermal conductivity and dynamic viscosity of the condensate respectively, evaluated at the condensate film temperature $T_{\text{film}} = (T_{\text{steam}} + T_{\text{surf}})/2 = T_{\text{steam}} - \Delta T/2$ of the subcooling. For $T_{\text{steam}} = 24.1$ °C (saturation temperature at 30 mbar) and $0.5 \text{ K} \leq \Delta T \leq 12 \text{ K}$, the range of values at T_{film} are:

$$\begin{aligned} 23.829 \text{ }^\circ\text{C} &\geq T_{\text{film}} \geq 18.079 \text{ }^\circ\text{C} \\ 997.30 \text{ kg m}^{-3} &\leq \rho \leq 998.54 \text{ kg m}^{-3} \\ 0.60453 \text{ W m}^{-1} \text{ K}^{-1} &\geq k \geq 0.59450 \text{ W m}^{-1} \text{ K}^{-1} \\ 9.1431 \times 10^{-4} \text{ kg m}^{-1} \text{ s}^{-1} &\leq \mu \leq 1.0506 \times 10^{-3} \text{ kg m}^{-1} \text{ s}^{-1} \end{aligned}$$

Figure R3.4 plots the prediction with the Nusselt model onto the heat transfer measurements on superhydrophobic and pristine boehmite. The measurements on pristine boehmite, on which filmwise condensation occurs, match closely with the predictions of the model.

Figure R3.4: Heat transfer coefficients on superhydrophobic and pristine boehmite. Predictions from the Nusselt model match closely with the filmwise condensation measurements on pristine boehmite.

Therefore, our experimental setup and procedures are validated to accurately measure heat transfer coefficients. We have updated the manuscript (p. 11) to include the work of Peng et al.¹⁹ and **Figure 3a** (p. 15) to include the Nusselt model predictions:

Filmwise condensation measurements are validated against the Nusselt model, as commonly found in literature¹⁹ (**Supplementary Information S2**).

Figure 3: Heat transfer performance of roaming condensation. (a) Heat transfer coefficients h at steady state. Lines of constant heat flux q'' are shown in grey, from 25 to 275 kW m⁻² at intervals of 50. For a fair test, the 7 subcooling achieved for each surface correspond to 7 identical cooler back end temperatures. The achievable subcooling range is smaller for a surface with better performance, as the surface temperature would closely approach the steam temperature in the same system. On the superhydrophobic surface, two modes of condensation are observed. **Measurements on pristine boehmite match closely with the Nusselt model for filmwise condensation.**

The explanatory text is added to **Supplementary Information S2** (p. 20):

Validation with the Nusselt model

The measurements for filmwise condensation are validated with the Nusselt model. For condensation from saturated steam on a small vertical plate, the model predicts the heat transfer coefficient to be:¹⁸

$$h = 1.13 \left[\frac{\rho(\rho - \rho_{\text{steam}})gh_{fg}k^3}{L\mu(T_{\text{steam}} - T_{\text{surf}})} \right]^{\frac{1}{4}}$$

where h_{fg} is the latent heat of vaporisation at 30 mbar (= 2443.9 kJ kg⁻¹). L is the length of the vertical condensing area in the gravitational direction (= 20 mm). ρ_{steam} is the density of the steam (= 0.021904 kg m⁻³). ρ , k and μ are the density, thermal conductivity and dynamic viscosity of the condensate respectively, evaluated at the condensate film temperature $T_{\text{film}} = (T_{\text{steam}} + T_{\text{surf}})/2 = T_{\text{steam}} - \Delta T/2$ of the subcooling. For $T_{\text{steam}} = 24.1$ °C (saturation temperature at 30 mbar) and $0.5 \text{ K} \leq \Delta T \leq 12 \text{ K}$, the range of values at T_{film} are:

$$\begin{aligned} 23.829 \text{ °C} &\geq T_{\text{film}} \geq 18.079 \text{ °C} \\ 997.30 \text{ kg m}^{-3} &\leq \rho \leq 998.54 \text{ kg m}^{-3} \\ 0.60453 \text{ W m}^{-1} \text{ K}^{-1} &\geq k \geq 0.59450 \text{ W m}^{-1} \text{ K}^{-1} \\ 9.1431 \times 10^{-4} \text{ kg m}^{-1} \text{ s}^{-1} &\leq \mu \leq 1.0506 \times 10^{-3} \text{ kg m}^{-1} \text{ s}^{-1} \end{aligned}$$

Figure 3a plots the prediction with the Nusselt model onto the heat transfer measurements on superhydrophobic and pristine boehmite. The measurements on pristine boehmite, on which filmwise condensation occurs, match closely with the predictions of the model. Therefore, our experimental setup and procedures are validated to accurately measure heat transfer coefficients.

3. Is the main mode of droplet removal still through gravity-driven shedding? It does not appear this was discussed.

An adequate answer to this question consists of two parts, as it depends on how droplet removal is defined. Roaming effectively removes microdroplets from their locations on the surface by speeding up coalescence compared to conventional droplet coalescence localised to a neighbourhood. This exposes the surface for another round of nucleation, and the emergence of much smaller droplet nuclei increases heat transfer efficiency due to their lower thermal resistance. On the other hand, for the condensate to be removed *from the surface* eventually, gravitational removal is the main mechanism. However, the high frequency of roaming events concurrently speeds up the attainment of the gravitational droplet departure diameter, i.e. droplets quickly grow and leave the surface in a shorter period of time compared to without roaming. We have discussed this in the manuscript (p. 12):

The entire process significantly speeds up the growth of droplets on the surface to attain the gravitational departure size (**Supplementary Information S9**). Eventually, the condensate droplets leave by gravity.

The frequent renewal of large surface areas (1) by roaming itself, and (2) from the increased gravitational departure assisted by roaming, enable abundant nucleation and ultimately maximise heat transfer.

In **Supplementary Information S9** (p. 42 and 43), we have provided more details:

Figure S9.2: Surface area renewal rate from gravitational motion of droplets. Unit: mm² of surface area renewed per second (left axis). Number of gravitational motion events observed. Unit: number per second (right axis). At ≈ 1.5 K, a sharp transition in slope in both measurements is seen.

The transition to roaming significantly increases the surface area renewal rate from gravitational motion as well. Refer to **Figure S9.2**. Crosses and the left axis show the surface area renewal rate from gravitational motion. Squares and the right axis show the number of observed gravitational events. At the transition subcooling of 1.5 K when roaming emerges, there is a sharp change in slope in both measurements. While the motion of roaming itself renews over 70% of the condensing area every second as indicated by S' , it also accelerates the gravitational departure of droplets and the surface area renewal rate from such departure. This is because roaming events often terminate with a droplet at rest on the surface, which markedly increases the growth rate of droplets on the surface: the serial motion in roaming intensifies coalescence frequencies. Droplets then attain the critical size for gravitational departure much

earlier than through only the direct condensation of vapour at a site, or local coalescence of sessile droplets in place, without roaming involved.

We have modified the manuscript (p. 13) and added **Figure 3e** (p. 15) to strengthen the presentation:

Figures 3d and 3e summarise the mechanism for roaming and its benefits to heat transfer, which we will discuss in detail in the upcoming sections.

Figure 3: (e) The high surface area renewal rate of roaming enables abundant renucleation. Frequent roaming also assists droplet growth to the required size of gravitational removal.

4. In Figure 3, the superhydrophobic surface is labeled as exhibiting jumping droplet condensation and roaming condensation as subcooling increases. Can the authors explain why the mechanism would start as jumping droplet condensation and then transition to roaming condensation? This transition is not quite clear. In Movie S5, it appears at low subcooling that the “jumping” resembles a faster version of roaming condensation. Can the authors clarify this observation?

The condensation mode begins as jumping dropwise condensation and transitions to roaming condensation at a higher subcooling. When subcooling is low, all of the nanostructure cavities are free of condensate because the nucleation diameter is above the characteristic length scale of the cavities. Thus, all condensate microdroplets remain in the Cassie state. When two such microdroplets coalesce, the resultant motion is predominantly normal to the surface as there is no asymmetry in adhesion. This manifests as droplet jumping.

However, at higher subcooling, the nucleation diameter reduces to that below the characteristic length scale of the nanostructures. Condensate nuclei begin to appear randomly in the nanostructure cavities, partially filling them. This results in some of the nanostructures being filled, while some remain clear of condensate. The microdroplets resting on top of partially filled nanostructures exhibits higher adhesion due to a partial Wenzel wetting state, and the microdroplets resting on top of nanostructures clear of condensate remain in the Cassie state with minimal adhesion. When two droplets with this asymmetry in adhesion to the surface coalesce, the release of excess surface energy results in a predominantly tangential momentum generation, leading to a roaming event. Therefore, roaming condensation is observed at higher subcooling levels than jumping dropwise condensation.

We have modified the manuscript (p. 13) and added **Figure 3d** (p. 15) for clarification:

Figures 3d and 3e summarise the mechanism for roaming and its benefits to heat transfer, which we will discuss in detail in the upcoming sections.

Figure 3: (d) When subcooling is increased past the transition, condensate nucleates within the nanostructures. Microdroplets on top of these nanostructures could then exhibit different wetting states. The asymmetric adhesion gives rise to substantial tangential momentum upon coalescence.

Refer to the snapshot of **Video S5** in **Figure R3.5**. On the left, there is jumping dropwise condensation. Here, all the droplets that are away from the surface have jumped. They travel over the droplets resting on the surface. These droplets that have jumped away appear darker, and out of focus. Due to a steam flow from right to left, these droplets which have jumped into the vapour follow the leftward steam flow and downward gravity, and travel to the bottom left of the field of view.

On the right, there is roaming condensation. Here, droplets roam in-plane on the surface. After each roaming event, there are no droplets remaining along the trajectory. The main droplet of a roaming event travels in-plane on the surface and coalesces with droplets along the path. Immediately after a roaming event, the trajectory becomes brighter than the background, as the absence of condensate droplets reduces scattering of light.

Figure R3.5: Snapshot of **Video S5** at 23 s with annotations. Jumping dropwise condensation is on the left. Droplets which have jumped are away from the surface, and they appear darker and out of focus. Some of them are indicated by red arrows. In the video, these darker and out-of-focus droplets travel quickly to the bottom left due to steam flow (leftward) and gravity (downward), see **Video S5**. On the right, there is roaming condensation. The approximate trajectory of some roaming events are indicated by blue arrows. Along the trajectory, there are no droplets left on the surface. As the main droplet roams, it coalesces with the droplets along its path.

In jumping dropwise condensation, there are numerous droplets travelling in the vapour, but they are in one general direction due to steam flow and gravity. In roaming condensation, the droplets remain on the surface, and the roaming trajectories are in all in-plane directions. Jumping dropwise condensation is not a “faster” version of roaming condensation. The jumped droplets appear “fast” because they are quickly carried away in the steam flow, but they are not on the surface and have no effect on the condensation efficiency.

We have modified **Figure 3b** (p. 15 of the manuscript) for better presentation:

Figure 3: (b) Snapshots of condensation behaviour for superhydrophobic boehmite. Transition is seen from jumping dropwise to roaming condensation. At the lowest subcooling (0.7 K), only jumping is observed and there are numerous droplets in the vapour, with darker appearance and out-of-focus contour. These droplets in the vapour travel in one general direction to the bottom left due to steam flow (leftward) and gravity (downward). At 1.3 K, the number of jumped droplets in the vapour is visibly reduced, and some are seen to return to the surface. After the transition subcooling (1.5 K), condensation is dominated by roaming. Red arrows are trajectories of roaming events. Roaming droplets travel in all in-plane directions. See **Video S5** for the corresponding video. Scale bars: 500 μm .

5. It is unclear what additional information the simulation results in Figures 4 and 5 provide that is not already shown in the experimental videos. Droplet coalescence events and the displacement of droplets can be easily observed in the videos.

In the experiments, we have established that roaming can occur from the adhesion asymmetry of droplets. Simulations complement the experimental observations with additional details on the coalescence mechanism between two droplets of different wetting states and show that tangential momentum is generated in such events.

In **Figure 4**, we study a case in which there are two isolated droplets. One of the droplets has a wetted base contact area with the substrate to reproduce adhesion asymmetry. **Figure 4b** shows the evolution of the two-droplet system at selected time points, and how tangential momentum is generated through asymmetric recoil (Panel ii) and asymmetric curvature (Panel iii). This mechanism results in significant tangential momentum and displacement, as we show in the top plot of **Figure 4c**, compared to the case in which both droplets are in the Cassie state without adhesion asymmetry (D1 D2 Cassie). Finally, in **Figure 4d**, we vary the size of the droplet wetted base area to show that there is a threshold after which tangential momentum generation is substantial. Only 20% of the droplet base area has to be wetted.

Simulations in **Figures 5a** to **5c(i)** focus on the dewetting after coalescence for the droplets to be able to roam. The proportion of tangential component in the resultant motion depends on the exact moment when dewetting occurs. **Figure 5b** shows the momentum evolution upon dewetting. Then, in **Figure 5c**, we show that this coalescence mechanism is more efficient in converting excess surface energy to kinetic energy, than what it would otherwise be in the conventional jumping case where two droplets are in the Cassie state. This is because when one droplet is wetted, the droplet system remains close to the surface, allowing stronger symmetry breaking over several cycles of oscillation.

We have modified the manuscript (p. 22) and added **Figure 5c(ii)** (p. 24) to strengthen our presentation:

The adhesion keeps the liquid body close to the surface, resulting in stronger and more instances of symmetry breaking than droplet jumping (**Figure 5c(ii)**).

Figure 5: (c)(ii) Schematic illustrating the symmetry breaking in jumping and roaming motions. When two droplets coalesce, the liquid body oscillates (numbered 1 – 4, 3 omitted in roaming for clarity). The interference with the surface breaks symmetry and generates momentum. In jumping, the liquid body leaves early and oscillates in the vapour. In roaming, the liquid body remains close to the surface. Oscillations interfere stronger with the surface repeatedly. The hinge then converts the symmetry breaking in the normal direction to a tangential direction.

The experiments in **Figures 1 to 3** cannot yield the insights above. Currently available experimental methods cannot probe and image the wetting state of individual microdroplets and their effect on momentum generation during coalescence, with the required temporal and spatial resolution during condensation from saturated steam. Numerical simulations enable understanding of the coalescence mechanism, and most importantly, the mechanism of tangential momentum generation.

It is also confusing why a no-slip boundary condition was used in the simulation, but then a 2° contact angle at the base of the droplet is applied. How does this contact angle interact with the base if there is a no-slip boundary condition?

For simulations of droplet behaviour on superhydrophobic surfaces, it is common practice^{20–25} to model the superhydrophobic surfaces as a no-slip wall, and specify apparent contact angles to model the droplet adhesion to the wall. In our simulations, the contact angle is generally set to be 160° . To model the increased adhesion of partially wetted droplets, the contact angle at the base of such droplets is set to 2° .

We choose the volume of fluid (VOF) method to model the coalescence process. Surface tension and wall adhesion are modelled with the continuum surface force (CSF) model by Brackbill et al.²⁶. Both are natively implemented in ANSYS Fluent.²⁷ The use of VOF^{20,22–25,28} and CSF^{20,22–25} is common in droplet simulations.

Under the VOF model, the entire flow field is solved from one momentum equation shared by the two (liquid water and water vapour) phases. The fluid properties of a cell are the volumetric mean of the phases in the cell. This is tracked by the volume fraction, typically denoted as α . Naturally, the liquid water volume fraction is 1 in the bulk of the water droplet, and 0 in the bulk of the water vapour region. At the liquid-vapour interface, the volume fraction is between 0 and 1, i.e. the interface is somewhere within the cell. To reconstruct the interface, a geometric reconstruction scheme²⁷ is adopted. The interface is represented as a piecewise linear surface, which is also used for the calculation of fluxes across cell faces.²⁷ The reconstruction step maintains a sharp interface across the computation. As the interface moves across time steps, to avoid numerical issues, in our implementation we automatically set the volume fraction of a cell to 0 when it drops below a cutoff of 10^{-6} , and to 1 when it increases to a cutoff of $1 - 10^{-6}$.

Surface tension is handled by the CSF model.²⁶ This model includes effects from surface tension in the momentum equation as a source, i.e. a force.²⁷ Refer to Refs.^{26,27} for detailed mathematical formulation. A contact angle can be specified as a boundary condition at the wall. The difference between this contact angle that the fluids are supposed to make with the wall in equilibrium, and the actual angle from the flow solution, forms a “local curvature” which imposes a force to the flow field through the CSF model.²⁷ The specified contact angle is then enforced.

In summary, the no-slip boundary condition applies to the momentum equation only, for the computation of the flow field. The droplet interface, on the other hand, is reconstructed from the distribution of the volume fraction in the flow field.²⁷ The contact angle boundary condition is applied through the CSF model at the contact line. Therefore there is no incompatibility in specifying both no-slip and contact angle boundary conditions at the surface. Rather, the no-slip boundary condition is commonly adopted together with specified contact angles.^{20–25}

Furthermore, the simulation clearly shows droplet jumping in Figure 5, so why is this referred to as roaming? At least, roaming is not a controllable phenomenon as jumping is inevitable. Recently, a coarsening droplet is reported to significantly enhance condensation in a controlled way (10.1002/dro2.23)

The simulation case in **Figure 5a** is mainly to illustrate how tangential momentum can remain in droplet motion after dewetting. In roaming, the droplet can detach at different angles (the arrows in **Figure 5a(iii)**), but it depends on the exact moment of dewetting (**Figure S14.1b**). Limited by currently available experimental methods, this exact moment cannot be known because it is dependent on the level of wetting of the partial-Wenzel droplet.

Nevertheless, the case in **Figure 5a** is still referred to roaming because there is substantial tangential component. In current literature, jumping motion is at most a few degrees off the normal of the surface it jumps from. For example, see Figure 2d in Ref.²⁹, which is also verified with our simulations in **Supplementary Information S12**. The defining feature of roaming is the large proportion of tangential component in droplet motion, such that there is an overall in-plane motion spanning across considerable distance. In **Figure 5a**, the droplet centre of mass is 56.6° from the positive y-axis at the initial centre of mass. This large angle can never be achieved in conventional droplet jumping.

The droplet may temporarily lose contact with the surface after dewetting due to a small normal momentum. However, the high nucleation rates during roaming condensation ensure the abundance of droplets on the surface. For most of the time, the droplet departing at such a large angle is intercepted by these droplets on the surface and roaming propagates. This progression is seen in **Figure 5d**, where there is a change in main droplet optical reflection when it is lifted from the surface slightly during dewetting. After it is intercepted by droplets on the surface, in-plane roaming continues.

We have added further discussion in **Supplementary Information S14** (p. 61):

The exact dewetting time is dependent on the adhesion strength, but a general feature is that significant tangential momentum is generated to result in a much higher angle of motion than the 4° deviation from surface normal in conventional droplet jumping without adhesion asymmetry (**Supplementary Information S12**). In **Figure 5a**, it is 56.6°. In rare cases, if the main droplet is not intercepted by droplets on the surface after dewetting, in-plane roaming is prevented and it could depart the surface with motion at these large angles.

Lastly, this loss of contact with the surface is not “inevitable”. As we show in **Figure S14.1b**, the droplet can remain fully in contact with the surface when the time of dewetting is slightly delayed. During experiments, we have also occasionally observed roaming events in which the main droplet travels without change in optical reflection, remaining in continuous contact with the surface.

We have included the work of Guo et al. to highlight the benefits brought by coarsening droplets to heat transfer enhancement:

The pursuit for efficient heat removal has strongly motivated surface engineering research, with the central idea to minimise the residence time and amount of the liquid condensate on the cooled surface.¹⁰

6. In the supporting movies S3 and S4, the phenomena described in the titles are not directly shown. In Movie S3, “Roaming events altering direction,” the roaming droplet simply moves to the left and never changes direction. In Movie S4, “Roaming events terminating in jumping,” the video never shows a droplet jumping at the end of roaming. If it does, it is not clear. These videos are misleading as they do not show the supposed behavior described in the manuscript.

We hope we can provide further clarification to **Videos S3** and **S4** in the following.

Video S3

For most of the roaming events that we have observed in this study, the main droplet travels in a relatively straight line. However, it is possible for them to travel along a curve in some cases. **Video S3** is intended to show these curved trajectories, in that both Clips 1 and 2 exhibit sharp turns. We have modified **Video S3** to show the trajectory explicitly. Refer to the snapshots of **Video S3** in **Figure R3.6**. The event in Clip 1 turns left and the event in Clip 2 turns right, while travelling in-plane on the surface.

Figure R3.6: Snapshots of the modified **Video S3**. (a) Clip 1 at 21 s, and (b) Clip 2 at 43 s.

We have added clarification to the manuscript (p. 7):

Occasionally, a roaming event can alter its direction as it progresses, resulting in an in-plane curved trajectory (Video S3).

Video S4

In **Video S4**, main droplets at the end of roaming depart the surface by jumping, with a magnitude enough to “jump over” and not coalesce with further droplets on the surface. We have modified **Video S4** to explicitly point out when the main droplets leave the surface and the droplets on the surface that the main droplets pass over. Refer to the snapshots of Clip 4 of **Video S4** in **Figure R3.7**. When the main droplet is lifted slightly from the surface, it becomes brighter for a short time. Then, it becomes darker and out of focus as it moves further away from the surface. The clear evidence of jumping can be seen when the droplet passes over the other droplets without coalescing. The roaming event has terminated in jumping of the main droplet.

Figure R3.7: Snapshots of Clip 4 of the modified **Video S4**, around the moment when roaming terminates in jumping. (a) The main droplet becomes brighter when it is lifted slightly from the surface. (b) The main droplet then becomes dark and out of focus as it has jumped further away from the surface. (c) The main droplet passes over droplets on the surface without coalescence.

7. Regarding the heat transfer comparison made in Figure S8.2, it is great that PDMS brushes were compared to the current work, as these are state-of-the-art. Ideally, it would be best to compare results taken from the same experimental set-up, but it is understandable that this is challenging. However, it may not be a fair comparison based solely on subcooling. It may be good to compare the heat transfer coefficient versus heat flux as well.

The PDMS brushes in **Figure S8.2** are tested in the same experimental setup under the same conditions. Therefore, it is a fair comparison to overlay the PDMS brushes data in **Figure S8.2**. We have added the following text to **Supplementary Information S8** (p. 39) to make it clear and added an additional plot of heat transfer coefficient against heat flux in **Figure S8.3** (p. 40 of Supplementary Information), as suggested by the reviewer:

As an example, **Figure S8.2** overlays the heat transfer coefficient measurements for a surface composed of polydimethylsiloxane (PDMS) brushes under the same experimental conditions in the same experimental setup¹⁶ on the plot in **Figure 3a**. **Figure S8.3** plots the heat transfer coefficients against heat flux.

Figure S8.3: Heat transfer measurements in **Figure S8.2**, with heat flux on the x-axis. Data for PDMS brushes from Ref.¹⁶. Lines of constant subcooling ΔT are shown in grey, from 0.5 to 11.5 K at intervals of 1.

8. In the current study, subcooling is defined as $\Delta T = T_{\text{steam}} - T_{\text{surf}}$. The term T_{steam} should be replaced with T_{sat} , where the value of subcooling is directly defined based on the saturation temperature of steam, depending on the thermodynamic state of the chamber conditions.

During our experiments, we verify that the steam is always at saturation ($T_{\text{steam}} = T_{\text{sat}}$) with pressure measurements and steam tables. Therefore, both definitions of subcooling would be equivalent whether the steam or saturation temperature is chosen. We have added the following to the manuscript (p. 5) to make it clear:

We observe the condensation behaviour as we increase the surface subcooling $\Delta T = T_{\text{steam}} - T_{\text{surf}}$, the difference between the steam temperature T_{steam} and the surface temperature T_{surf} . As the steam is always at saturation, T_{steam} equals the saturation temperature at our operating pressure (T_{sat} at 30 mbar).

9. The supporting videos are too large. There are ways to reduce the size of each video significantly while maintaining the resolution.

The videos in Supplementary Information are now further compressed to smaller file sizes.

REVIEWER #4:

In this work, the authors report the in-plane roaming of condensate droplets upon coalescence on nanostructured superhydrophobic surfaces. Despite the authors' claims, the novelty and significance of this work are questionable. The authors need to address the below comments before the manuscript can be accepted for publication.

We thank the reviewer for the time and effort in reviewing our manuscript. We hope the following responses can provide satisfactory clarification to our work for its reconsideration.

1. The authors themselves mention and cite references that tangential motion of condensate droplets has been well-studied previously on microstructured surfaces. Here, the authors report a similar phenomenon on nanostructured surfaces, terming it "roaming droplets." Thus, the work has limited novelty and uniqueness.

The fundamental novelty and uniqueness of our work lies in the fact that we show that the tangential motion of coalescing condensate droplets *does not require microstructures at all*. We present an original mechanism through which condensate droplets can roam upon coalescence, namely the asymmetry in adhesion due to stochastic nucleation within nanostructures. In existing literature, tangential droplet motion upon coalescence is explained by jumping on the side walls of the microstructure cavities.²⁹⁻³³ This explanation becomes invalid for tangential motion observed on solely nanostructured surfaces as there are no microstructures to jump laterally from. Therefore, we have developed a completely new theory in this work to explain such a phenomenon. It is neither fair nor correct to claim that the two phenomena are "similar," as they rely on completely different physics.

Our work is the first in at least the following facts:

1. Identifying roaming motion of coalescing condensate microdroplets on solely nanostructured superhydrophobic surfaces, which indicates a different tangential momentum generation mechanism from the ones previously understood;
2. Observing evolving roaming motion with sufficient temporal and spatial resolution for characterisation from initiation to termination;
3. Quantitatively characterising roaming including propagation velocities and event lengths;
4. Simultaneous in-situ measurement of heat transfer, showing that roaming condensation provides the highest heat transfer coefficient, even higher than jumping dropwise condensation; and
5. Providing a mechanism for tangential momentum generation arising from the spontaneous wettability asymmetry when condensation occurs stochastically within nanostructure cavities. Currently available studies have only pointed to increased adhesion, which is itself insufficient for producing tangential motion.

Therefore, we believe this work is novel and unique in the field, sharing no overlap with studies currently available in literature.

2. Despite the detailed discussion, it is not clear why in-plane roaming condensation cannot occur at a lower degree of subcooling. Even at lower subcooling, the probability of droplets jumping upon coalescence is low, and the asymmetry during coalescence still exists. Why is there no roaming effect at lower subcooling?

We do not agree that the probability of droplets jumping upon coalescence is low. On a superhydrophobic surface, when two Cassie state droplets coalesce, they almost always jump and depart the surface, as long as they are in the diameter range in which capillary effects dominate. When subcooling is low, there is no significant asymmetry in *adhesion* for the condensing droplets. This is because the droplets are in Cassie state and there is minimal adhesion for all droplets anyway. However, when subcooling increases, condensate can start nucleating within the nanostructure cavities as the critical nucleation diameter decreases. This alters the wetting state of the droplets on top of these wetted nanostructures, from Cassie to a partial-Wenzel state. Since nucleation is stochastic, at the required subcooling range, some of these droplets are in the altered state (partial-Wenzel) while some remain in

the Cassie state. The large asymmetry of adhesion between the two wetting states provides a hinge for tangential momentum generation as two droplets of different adhesion to the surface coalesce. Since there cannot be nucleation within the nanostructure cavities at lower subcooling, asymmetry in adhesion is absent and consequently no roaming motion can be generated.

We have modified the manuscript (p. 13) and added **Figure 3d** (p. 15) to strengthen our presentation:

Figures 3d and 3e summarise the mechanism for roaming and its benefits to heat transfer, which we will discuss in detail in the upcoming sections.

Figure 3: (d) When subcooling is increased past the transition, condensate nucleates within the nanostructures. Microdroplets on top of these nanostructures could then exhibit different wetting states. The asymmetric adhesion gives rise to substantial tangential momentum upon coalescence.

3. The discussion on the limitations of roaming condensation is significantly lacking. Jumping droplet condensation can occur on a wide range of surfaces (nanostructured, hierarchical, etc.). However, roaming condensation seems to be possible solely on nanostructured surfaces with unique surface morphologies.

In our experiments, we have observed tangential motion on hierarchically superhydrophobic surfaces as well. Microstructures are imparted onto aluminium substrates by etching with iron(III) chloride, before nanostructuring. The resulting hierarchical micro-nano structures can be seen in **Figure R4.1**. A droplet can be seen travelling in-plane on this surface in **Figure R4.2**.

Figure R4.1: Hierarchically structured aluminium. The microstructures are overlaid with nanostructures. Scale bar: 5 μm .

Figure R4.2: Droplet travelling in-plane on the hierarchically structured surface. White arrow points to the main droplet. Red arrow indicates approximate trajectory. Scale bar: 100 μm .

However, we have chosen to focus primarily on nanostructured superhydrophobic surfaces in this work to eliminate the effects from microstructures. In particular, any apparent tangential droplet motion from the jumping of droplets on the side walls of microstructure cavities is no longer possible on a solely nanostructured surface. All tangential motion has to therefore result from roaming effects and asymmetry in droplet adhesion, as we show and explain in this work. When microstructures are present, the effect of (1) asymmetric adhesion and (2) jumping from sidewalls cannot be decoupled. We have added the following text towards the end of the manuscript (p. 26) to note the possibility of tangential motion on hierarchical surfaces:

In this work we have focused on solely nanostructured superhydrophobic surfaces to eliminate the effects from microstructures. However, roaming on hierarchical surfaces might be possible (**Supplementary Information S16**).

We also added the following text as **Supplementary Information S16** (p. 70) as elaboration:

Roaming events shown in this work are on solely nanostructured surfaces, namely superhydrophobic boehmite nanowalls, copper(II) hydroxide nanoneedles and titanium dioxide nanorods. This has eliminated effects from microstructures. In particular, any apparent tangential droplet motion from the jumping of droplets on the side walls of microstructure cavities is not possible. All tangential motion has to therefore result from roaming effects and asymmetry in droplet adhesion, as we have shown and explained.

Nevertheless, roaming on hierarchical surfaces might be possible. In **Figure S16.1**, a hierarchical aluminium surface is shown. Microstructures are imparted onto the substrate by etching with iron(III) chloride, before nanostructuring. A droplet travelling in-plane can be seen in **Figure S16.2**.

Figure S16.1: Hierarchically structured aluminium. The microstructures are overlaid with nanostructures. Scale bar: 5 μm .

Figure S16.2: Droplet travelling in-plane on the hierarchically structured surface. White arrow points to the main droplet. Red arrow indicates approximate trajectory. Scale bar: 100 μm .

The event in **Figure S16.2** resembles very closely the roaming behaviour we observe on solely nanostructured surfaces. However, the presence of microstructures interferes with the interpretation because it is no longer possible to decouple the effects from (1) asymmetric adhesion and (2) jumping from side walls. The tangential momentum observed may be a result of both. Therefore, in our work, we have focused exclusively on solely nanostructured surfaces.

Although solely nanostructured superhydrophobic surfaces are the focus of our work, we clearly disagree that roaming can only occur on those with unique surface morphologies. First, while surfaces can be made superhydrophobic for sessile droplets through the introduction of structures, only a few superhydrophobic surfaces are somewhat resistant to immediate flooding in condensation of saturated steam. This is because the structures have to possess some ability to repel droplets nucleated within them to maintain the vapour layer. Second, substrates considered should be thermally conductive, most likely metallic, for heat transfer purposes. The choice of superhydrophobic surface should therefore be viewed in the context of this subset of available surfaces. In our work, we have shown that roaming occurs on at least three types of nanostructures, namely boehmite nanowalls, copper(II) hydroxide nanoneedles, and titanium dioxide nanorods (**Supplementary Information S5**).

The aluminium, copper and titanium-based nanostructures studied in our present work differ by at least:

1. Characteristic structure size;
2. Whether the nanostructure cavities are open or closed;
3. Sparsity of the nanostructure cavities; and
4. Shape of the nanostructure features.

Figure R4.3 displays the three types of nanostructures next to each other. They differ much from each other in the above criteria.

Figure R4.3: SEM images of titanium dioxide nanorods (top), copper(II) hydroxide nanoneedles (middle) and boehmite nanowalls (bottom). Scale bar: 2 μm . The surface morphologies differ significantly in terms of size, cavity openness, and cavity sparsity.

All surface morphologies are unique by themselves, and we have selected a few with significant differences from each other to show that roaming is independent of the form of nanostructures. We believe the phenomenon of roaming is universal on superhydrophobic surfaces, given that they can sustain condensation from saturated vapour without flooding immediately.

4. The current work focuses on condensation on superhydrophobic surfaces. However, the introduction discusses lubricant-infused surfaces at length and cites several references, which is unwarranted.

The removal of droplets and the mechanisms for their spontaneous motion underlies much of condensation heat transfer research, in that they are critical in exposing new surface area for nucleation to minimise heat transfer resistance. Therefore, we have cited a few references in the first paragraph to provide a broader background on how this has become an important topic in the heat transfer area. Within this context, we show in this work that roaming of condensate droplets maximises heat transfer efficiency with nanostructured superhydrophobic surfaces. We have not discussed lubricant-infused surfaces in the manuscript.

5. The authors emphasize the higher heat transfer coefficient (HTC) for roaming condensation versus jumping droplet condensation. This is not a fair comparison. Generally, the heat transfer coefficient increases with increasing subcooling during dropwise condensation. Roaming condensation is realized only at higher subcooling, while jumping droplet condensation occurs at lower subcooling. How do the authors decouple the effects of subcooling degree and condensation mode on HTC?

For condensation modes in which the condensate manifests as discrete droplets, heat transfer coefficient does not always increase with subcooling. The increase in heat transfer coefficient with subcooling depends on whether the higher nucleation rates and smaller nucleation diameters at higher subcooling can be sustained by the droplet removal rate. There is no general trend, for example, the opposite (decreasing heat transfer coefficient with increasing subcooling) occurs in Ref.¹⁷.

From our experience, we have observed heat transfer coefficients weakly decreasing with increasing subcooling for a (gravitational) dropwise condensing polytetrafluoroethylene before (PTFE in Figure 3a in Ref.¹⁵, same experimental setup as current work). On the other hand, for surfaces with better droplet removal abilities, we have observed increasing heat transfer coefficients with increasing subcooling for at least a range of subcooling. For example, on PDMS brushes (Figure 3b in Ref.¹⁶, same experimental setup as current work) and lubricant-infused surfaces (VG LIS in Figure 3a in Ref.⁵, same experimental setup as current work). Therefore, for condensation modes other than filmwise condensation (i.e. there are discrete droplets), the trend of heat transfer coefficient with subcooling is highly dependent on the surface type, mostly on its ability to remove droplets efficiently.

We have modified slightly the manuscript (p. 13) for a more precise presentation:

The heat transfer coefficient can exhibit different trends with subcooling for surfaces on which condensate appears in the form of droplets.^{5,15-17} When droplet removal is sufficiently efficient, it can increase with subcooling,^{5,16} due to higher nucleation rates and smaller nucleation diameters. This important trend is also present in the superhydrophobic boehmite heat transfer coefficients in **Figure 3a**. However, this trend sustains so long as the condensation mode allows it, that is, there is limited flooding of structures or saturation of nucleation sites because of the efficient droplet removal. The > 70% area renewed every second in roaming condensation enables ample space for fresh nucleation, sustaining the trend.

In our work, it is shown that roaming condensation can indeed sustain the increase in heat transfer coefficient with subcooling after jumping subsides, because of its high frequency of events.

Secondly, it is not possible, on a given surface, to decouple

- (1) the relationship between heat transfer coefficients and subcooling; and
- (2) the relationship between condensation behaviour (mode) and subcooling.

When a given surface is subject to a given subcooling, there is a corresponding condensation behaviour, and its heat transfer efficiency is measured by the heat transfer coefficient. The transition in condensation mode, the increase in heat transfer coefficient, and the increase in subcooling are all coupled with one another. It is not representative of the physics of the problem to suggest that the heat transfer coefficient would increase with subcooling even without roaming, as it is not possible – roaming is bound to occur at that subcooling for a given surface.

In other words, jumping dropwise condensation cannot be achieved on the same surface at the same subcooling. At that subcooling, roaming condensation must occur, resulting in the corresponding high heat transfer coefficients.

The same argumentation can be employed in comparing other condensation modes. A hydrophobic surface can be made superhydrophobic by the introduction of structures. At a certain subcooling, jumping dropwise condensation

occurs on the superhydrophobic surface and conventional gravitational dropwise condensation occurs on the hydrophobic surface. Here, an experimentalist asserts that the heat transfer coefficient of jumping dropwise condensation is higher than that of conventional dropwise condensation, but cannot decouple the result from the change of the condensing surface itself – structuring increases heat transfer area, and alters the energetic barrier of nucleation due to curvature. Similarly, a hydrophilic surface can be made hydrophobic by altering the surface chemistry. At a certain subcooling, conventional dropwise condensation occurs on the hydrophobic surface and filmwise condensation occurs on the hydrophilic surface. The experimentalist asserts that the heat transfer coefficient of conventional dropwise condensation is higher than that of filmwise condensation, but again cannot decouple the result from the change of the condensing surface itself. In these two cases, it is the change of surface that enables a new condensation mode at the same subcooling, and in turn enables the corresponding improved heat transfer coefficients. In our case, it is the change of subcooling that enables the new condensation mode (roaming) on the same surface, and in turn enables the corresponding higher heat transfer coefficients.

Therefore, either the surface or subcooling has to be changed to effect another condensation mode and its corresponding heat transfer coefficients. It is just as fair to compare different condensation modes because of a change in subcooling (on a given surface), as to compare different condensation modes because of a change of the surface (at a given subcooling), as long as the context is clear.

6. In the “generation of tangential momentum in roaming” section, the authors state “without this, there is no apparent reason for out-of-plane jumping to in-plane roaming.” This statement is speculative and lacks sufficient explanation and support.

There are a few proposed mechanisms for spontaneous droplet motion in literature on different types of surfaces. In our work, we have identified the relevant ones on superhydrophobic surfaces and explained why they do not apply to droplet roaming. We therefore believe that the mechanism we proposed is sufficient to explain roaming motion upon droplet coalescence.

Conventionally, it is understood that tangential droplet motion on a superhydrophobic surface can be explained by droplet jumping – that the droplet of a length scale similar to surface microstructures jumps from the side walls of the microstructure cavities. We point this out as follows in the manuscript (p. 3):

It has been postulated that,²⁹⁻³³ as the condensate droplets are at the same length scale as the individual microfeatures, coalescence on the side walls of the microstructure cavities triggers inclined jumping in random directions.

And in **Supplementary Information S3** (p. 23):

Figure S3.1: Different droplet motion behaviour. (a) Motion from droplet jumping is normal to the surface. (b) When there are microstructures which are at the same length scale as the droplets, the direction of jumping can be altered. (c) Roaming can occur without microstructures. **a** and **c** have appeared on <https://doi.org/10.1103/APS.DFD.2023.GFM.V0073> in the 2023 American Physical Society Division of Fluid Dynamics Gallery of Fluid Motion.

Coalescence-induced jumping of microdroplets is normal to the surface (**Figure S3.1a**). Therefore, when it occurs on the lateral walls of microstructures, the resultant jumping motion would be sideways,³¹ as shown in **Figure S3.1b**.

The mechanism for roaming motion is entirely different, in that it can occur on solely nanostructured surfaces, producing a planar motion without the presence of any microstructures (**Figure S3.1c**). The planar (tangential) motion is inherent to the phenomenon itself.

In short, because there are no microstructures, the conventional explanation for tangential momentum generation, i.e. due to jumping from side walls, cannot apply to roaming. The nanostructures are orders of magnitude smaller than the droplets in concern.

Next, we point out that mere increase in adhesion of droplets to the surface would not produce tangential momentum. We explain in the manuscript (p. 17):

If the increase in adhesion for all condensate droplets on the surface were uniform when subcooling increases, i.e. no asymmetry, jumping would gradually cease and transition to flooded condensation would be directly evident without any intermediate in-plane roaming regime.

Therefore, there has to be an asymmetry in adhesion, which is reasonably provided by the stochastic nature of nucleation, occurring when we are in the subcooling regime, that nuclei can begin to form within the nanostructure cavities.

Last, we point out that substantial tangential momentum cannot be generated by coalescence of droplets with size mismatch. Simulations (**Supplementary Information S12**) show that for cases with size mismatches that result in droplet jumping, the maximum deviation from surface normal is $< 4^\circ$, consistent with literature.²⁹ For cases

with size mismatches that do not result in droplet jumping, the maximum centre-of-mass displacement in the tangential direction in 2 ms is less than 1% of the diameter of the smaller droplet.

We believe we have exhaustively shown that the mechanisms considered most plausible in the current literature cannot explain the roaming phenomenon.

7. In-plane roaming condensation should induce a substantial sweeping effect, clearing out condensate droplets leading to rapid regeneration of nucleation sites. However, there is hardly any discussion of this effect.

We agree with the reviewer that in-plane roaming sweeps and renews the surface rapidly. This increases heat transfer efficiency because the renewed surface area is frequently made available again for nucleation. A large number of small condensate nuclei form and become responsible for a significant amount of heat flow due to their much smaller thermal resistance. In our work, we have quantified this “substantial sweeping effect” from roaming motion with the parameter surface area renewal rate S' . Two main observations have been made. Firstly, a sudden increase of S' at the subcooling of 1.5 K indicates a sharp transition to roaming condensation. Secondly, when roaming condensation is dominant, > 70% of the condensing surface is renewed every second. We have discussed this in the manuscript (p. 11):

We quantify the transition in the top subplot of **Figure 3c** and show that when the subcooling increases past the transition at ≈ 1.5 K, the surface area renewal rate S' from roaming sharply increases. Remarkably, over 70% of the surface is renewed every second by roaming when it is the dominant mode.

Droplets coalesce during a roaming event. Therefore, apart from the area renewed by roaming motion itself, the high velocity and frequency of roaming events enable droplets on a condensing surface to rapidly coalesce and grow. They attain the gravitational departure diameter much earlier than it would otherwise be on a surface without roaming. As a result, when roaming condensation becomes dominant, gravitational motion becomes more frequent as well, which brings its own additional sweeping (in the gravitational direction) and surface area renewal. We have quantified the surface area renewal rate from gravitational motion in **Supplementary Information S9** (p. 42 and 43):

Figure S9.2: Surface area renewal rate from gravitational motion of droplets. Unit: mm² of surface area renewed per second (left axis). Number of gravitational motion events observed. Unit: number per second (right axis). At ≈ 1.5 K, a sharp transition in slope in both measurements is seen.

The transition to roaming significantly increases the surface area renewal rate from gravitational motion as well. Refer to **Figure S9.2**. Crosses and the left axis show the surface area renewal rate from gravitational motion. Squares and the right axis show the number of observed gravitational events. At the transition subcooling of 1.5 K when roaming emerges, there is a sharp change in slope in both

measurements. While the motion of roaming itself renews over 70% of the condensing area every second as indicated by S' , it also accelerates the gravitational departure of droplets and the surface area renewal rate from such departure. This is because roaming events often terminate with a droplet at rest on the surface, which markedly increases the growth rate of droplets on the surface: the serial motion in roaming intensifies coalescence frequencies. Droplets then attain the critical size for gravitational departure much earlier than through only the direct condensation of vapour at a site, or local coalescence of sessile droplets in place, without roaming involved.

Therefore, not only does roaming itself provide a “substantial sweeping effect,” it also provides additional sweeping from the gravitational motion made more frequent by roaming. All this improves heat transfer efficiency, as we discuss in the manuscript (p. 12):

In condensation, most of the heat and mass transfer is attributed to the initial droplet growth after nucleation.¹⁴ On a surface with a distribution of various condensate droplet sizes, the majority of heat flows through the smallest droplets. When subcooling increases, (1) the diameter at which nucleation occurs, i.e. the critical nucleation diameter d_{crit} , decreases, and (2) the nucleation rate (number of nuclei per area per time) increases.¹³ Therefore, at elevated subcooling, a renewed surface area is soon filled with a large number of small condensate droplets ideal for heat transfer. The frequent renewal of large surface areas (1) by roaming itself, and (2) from the increased gravitational departure assisted by roaming, enable abundant nucleation and ultimately maximise heat transfer.

8. Condensation on superhydrophobic surfaces also induce hierarchical condensation, where smaller nucleating droplets within the surface structures are absorbed by the larger sink droplet on the surface. Can such events induce the roaming effect?

It is true that for structured superhydrophobic surfaces, condensate nucleated within the structure cavities can coalesce with and effectively be absorbed by the larger droplets on top of these structures. However, as subcooling increases, this increasingly becomes difficult and improbable, eventually resulting in droplets in different wetting states. The difference in wetting state implies asymmetry in adhesion, which forms the basis of roaming motion as we show in this work. Roaming dewets the structures, sustaining a stable condensation mode.

In the subcooling range where it does happen that the condensate within the cavities coalesces with the droplets on top, we do not expect roaming motion to result. The difference in the size of the coalescing droplets is so large that the capillary waves are quickly dampened before it can be converted to bulk droplet motion.

We have modified the manuscript (p. 18) as follows to include the above discussion:

While nuclei within the nanostructure cavities may coalesce with and effectively be absorbed by the microdroplets on top, we do not expect any substantial motion of the microdroplet to result from such coalescence. It is because the vast difference in size of the coalescing droplets quickly dampens capillary waves before they are converted to bulk droplet motion. Moreover, when subcooling increases, this becomes increasingly difficult and improbable due to the high nucleation rates within the structure cavities. Microdroplets with different wetting states will eventually form. Roaming is a consequence of the adhesion asymmetry of droplets under different wetting states, and propagates by the dewetting of the partial-Wenzel state droplets.

9. How would the roaming condensation dynamics change if the condensing surface were a tube instead of a flat surface?

We do not expect material change to the condensation and droplet dynamics if the surface was curved (as on a tube). In our experiments, the roaming events have a mean travelled distance of 744 μm . Unless the radius of curvature is on these scales, which translates to extremely small tubes, the path along which the roaming droplet

traverses would be effectively as if on a flat superhydrophobic surface. We have added the following to the manuscript (p. 26):

Roaming events in our work travel over distances below 1 mm, suggesting that roaming might also be possible on highly curved surfaces.

Additionally, if the surface morphology changes from nanostructured alone to hierarchical structures (e.g., microstructured bumps with nanoneedles), how would roaming condensation be affected?

As explained in our response to Comment #3, roaming condensation could occur on hierarchical superhydrophobic surfaces. We chose to exclude any microstructures in our study primarily for the fact that jumping on the side walls of microstructure cavities provides an additional and entirely different mechanism for tangential momentum generation to our proposed mechanism of asymmetric adhesion. These two mechanisms would have been difficult to decouple and analyse independently.

On hierarchically structured surfaces, this additional mechanism would be responsible for some tangential motion itself. We expect that roaming motion arising from asymmetric adhesion would still occur, but some of these events may be affected by tangential momentum generated through other means.³¹ Since they cannot be easily decoupled, all tangential motion is to be analysed together and there would be larger scatter in measurements of duration, length, and velocity of the tangential motion events. Regarding heat transfer, we speculate some improvement in efficiency, due to (1) the larger surface area for hierarchically structured surfaces than solely nanostructured surfaces, and (2) a potentially higher surface area renewal rate due to the additional tangential momentum generation mechanism. More nuclei can therefore form in a given period of time.

We have added the following to **Supplementary Information S16** (p. 71):

We speculate that roaming on hierarchical surfaces may provide further improvement in heat transfer efficiency, due to (1) the larger surface area for hierarchically structured surfaces than solely nanostructured surfaces, and (2) a potentially higher surface area renewal rate due to the additional tangential momentum generation mechanism. More nuclei can form in a given period of time.

REFERENCES

1. Zhu, J.-L., Shi, W.-Y., Wang, T.-S. & Feng, L. Spontaneous thermocapillary motion of condensation droplets. *Appl. Phys. Lett.* **116**, 243703 (2020).
2. Oh, J. *et al.* Thin Film Condensation on Nanostructured Surfaces. *Adv. Funct. Mater.* **28**, 1707000 (2018).
3. Tripathy, A. *et al.* Ultrathin Durable Organic Hydrophobic Coatings Enhancing Dropwise Condensation Heat Transfer. *Langmuir* **38**, 11296–11303 (2022).
4. Cha, H. *et al.* Dropwise condensation on solid hydrophilic surfaces. *Sci. Adv.* **6**, eaax0746 (2020).
5. Tripathy, A. *et al.* Ultrathin Lubricant-Infused Vertical Graphene Nanoscaffolds for High-Performance Dropwise Condensation. *ACS Nano* **15**, 14305–14315 (2021).
6. Liu, J. *et al.* One-Step Synthesis of a Durable and Liquid-Repellent Poly(dimethylsiloxane) Coating. *Adv. Mater.* **33**, 2100237 (2021).
7. Anand, S., Paxson, A. T., Dhiman, R., Smith, J. D. & Varanasi, K. K. Enhanced Condensation on Lubricant-Impregnated Nanotextured Surfaces. *ACS Nano* **6**, 10122–10129 (2012).
8. Paxson, A. T., Yagüe, J. L., Gleason, K. K. & Varanasi, K. K. Stable Dropwise Condensation for Enhancing Heat Transfer via the Initiated Chemical Vapor Deposition (iCVD) of Grafted Polymer Films. *Adv. Mater.* **26**, 418–423 (2014).
9. Miljkovic, N. *et al.* Jumping-Droplet-Enhanced Condensation on Scalable Superhydrophobic Nanostructured Surfaces. *Nano Lett.* **13**, 179–187 (2013).
10. Guo, Z., Monga, D., Shan, L., Boylan, D. & Dai, X. Coarsening-induced disappearing droplets contribute to condensation. *Droplet* **1**, 170–181 (2022).
11. Dorrer, C. & Rühle, J. Wetting of Silicon Nanograss: From Superhydrophilic to Superhydrophobic Surfaces. *Adv. Mater.* **20**, 159–163 (2008).
12. Wen, R. *et al.* Hierarchical Superhydrophobic Surfaces with Micropatterned Nanowire Arrays for High-Efficiency Jumping Droplet Condensation. *ACS Appl. Mater. Interfaces* **9**, 44911–44921 (2017).
13. Carey, V. P. *Liquid-Vapor Phase-Change Phenomena: An Introduction to the Thermophysics of Vaporization and Condensation Processes in Heat Transfer Equipment.* (CRC Press, Taylor & Francis Group, Boca Raton, 2020).
14. Kim, S. & Kim, K. J. Dropwise Condensation Modeling Suitable for Superhydrophobic Surfaces. *J. Heat Transf.* **133**, 081502 (2011).

15. Donati, M. *et al.* Sprayable Thin and Robust Carbon Nanofiber Composite Coating for Extreme Jumping Dropwise Condensation Performance. *Adv. Mater. Interfaces* **8**, 2001176 (2021).
16. Li, S. *et al.* Durable, Ultrathin, and Antifouling Polymer Brush Coating for Efficient Condensation Heat Transfer. *ACS Appl. Mater. Interfaces* **16**, 1941–1949 (2024).
17. Boylan, D., Monga, D., Shan, L., Guo, Z. & Dai, X. Pushing the Limit of Beetle-Inspired Condensation on Biphilic Quasi-Liquid Surfaces. *Adv. Funct. Mater.* **33**, 2211113 (2023).
18. Holman, J. P. *Heat Transfer*. (McGraw-Hill, New York, 2002).
19. Peng, B., Ma, X., Lan, Z., Xu, W. & Wen, R. Experimental investigation on steam condensation heat transfer enhancement with vertically patterned hydrophobic–hydrophilic hybrid surfaces. *Int. J. Heat Mass Transf.* **83**, 27–38 (2015).
20. Chu, F., Yuan, Z., Zhang, X. & Wu, X. Energy analysis of droplet jumping induced by multi-droplet coalescence: The influences of droplet number and droplet location. *Int. J. Heat Mass Transf.* **121**, 315–320 (2018).
21. Liu, F., Ghigliotti, G., Feng, J. J. & Chen, C.-H. Numerical simulations of self-propelled jumping upon drop coalescence on non-wetting surfaces. *J. Fluid Mech.* **752**, 39–65 (2014).
22. Ravi Annapragada, S., Murthy, J. Y. & Garimella, S. V. Prediction of droplet dynamics on an incline. *Int. J. Heat Mass Transf.* **55**, 1466–1474 (2012).
23. Ravi Annapragada, S., Murthy, J. Y. & Garimella, S. V. Droplet retention on an incline. *Int. J. Heat Mass Transf.* **55**, 1457–1465 (2012).
24. Li, S., Chu, F., Zhang, J., Brutin, D. & Wen, D. Droplet jumping induced by coalescence of a moving droplet and a static one: Effect of initial velocity. *Chem. Eng. Sci.* **211**, 115252 (2020).
25. Yuan, Z., Wu, R. & Wu, X. Numerical simulations of multi-hop jumping on superhydrophobic surfaces. *Int. J. Heat Mass Transf.* **135**, 345–353 (2019).
26. Brackbill, J. U., Kothe, D. B. & Zemach, C. A continuum method for modeling surface tension. *J. Comput. Phys.* **100**, 335–354 (1992).
27. ANSYS, Inc. *ANSYS Fluent Theory Guide Release 2021 R2*. (2021).
28. Chu, F., Li, S., Ni, Z. & Wen, D. Departure Velocity of Rolling Droplet Jumping. *Langmuir* **36**, 3713–3719 (2020).
29. Yan, X. *et al.* Droplet Jumping: Effects of Droplet Size, Surface Structure, Pinning, and Liquid Properties. *ACS Nano* **13**, 1309–1323 (2019).

30. Mohammadian, B. *et al.* Delayed Frost Growth on Nanoporous Microstructured Surfaces Utilizing Jumping and Sweeping Condensates. *Langmuir* **36**, 6635–6650 (2020).
31. Qu, X. *et al.* Self-propelled sweeping removal of dropwise condensate. *Appl. Phys. Lett.* **106**, 221601 (2015).
32. Rykaczewski, K. *et al.* Multimode Multidrop Serial Coalescence Effects during Condensation on Hierarchical Superhydrophobic Surfaces. *Langmuir* **29**, 881–891 (2013).
33. Zhang, P., Maeda, Y., Lv, F., Takata, Y. & Orejon, D. Enhanced Coalescence-Induced Droplet-Jumping on Nanostructured Superhydrophobic Surfaces in the Absence of Microstructures. *ACS Appl. Mater. Interfaces* **9**, 35391–35403 (2017).

Paper title: Condensate droplet roaming on nanostructured superhydrophobic surfaces

The original remarks by the reviewers are shown verbatim in blue in this document. Direct quotes from the manuscript are in green in this document. Major modifications made to the manuscript as a result of the responses to the reviewers are highlighted both in this document and in the revised documents.

REVIEWER #3:

The authors have addressed all my comments. I recommend it for publication and congratulate the authors for their publication.

We thank the reviewer for the time in reviewing the manuscript and the positive opinion.

REVIEWER #4:

The authors have addressed most of the queries raised, significantly enhancing the quality of the manuscript. However, a few additional clarifications would be beneficial.

We thank the reviewer for the time in reviewing the manuscript. We hope the following clarification and the corresponding revision of the manuscript would address any remaining concerns of the reviewer.

1) It is understood that, on a given surface, decoupling the relationships between heat transfer coefficients and subcooling, as well as between condensation behavior (mode) and subcooling, is challenging. The initial comment stemmed from this complexity. Specifically, the authors compare heat transfer coefficients in the jumping droplet regime to those in the roaming droplet regime, suggesting enhancement. This comparison warrants caution, as multiple factors could influence the results, and the enhancement may not solely be due to roaming droplets. Including a sentence or two in the discussion to address this nuance would be valuable.

We agree with the reviewer that multiple factors could be affecting the final measured heat transfer coefficients. When subcooling increases, the critical nucleation diameter decreases and the nucleation rate increases. In the manuscript, we have discussed the conditions under which roaming motion can arise. The change in condensation behaviour is manifested most prominently as the transition from the jumping dropwise mode to the roaming dropwise mode takes place. The accompanying continuous increase in heat transfer coefficient is therefore undeniably intimately connected to the emergence of roaming.

Nevertheless, it is also true that more active condensation at higher subcooling can introduce other collateral effects to the overall condensation process. For example, there is more frequent coalescence (roaming or localised) due to higher droplet densities. The coupled effects, together with the very clear and prominent roaming motion of droplets over the entire surface, constitute the overall roaming condensation mode which we term as roaming condensation. To address and clarify these nuances, we have added the following text to the manuscript (p. 12):

In addition, the higher condensation rates when subcooling increases, inevitably translate to more active condensation behaviour. For example, there is a higher droplet density and thus a higher frequency of (roaming or localised) coalescence events. Such collateral effects, together with the prominent roaming motion of droplets over the entire surface, piece together the overall roaming condensation mode with its high heat transfer coefficients.

2) Hierarchical condensation is observed across a broad range of droplet sizes (Yan et al., ACS nano 13.7 (2019): 8169-8184.), which includes those reported by the authors. Could the authors elaborate on why this phenomenon would not induce roaming droplets, beyond describing it as increasingly difficult and improbable?

In the previous review, the reviewer made a remark on the same matter which we quote verbatim:

Condensation on superhydrophobic surfaces also induce hierarchical condensation, where smaller nucleating droplets within the surface structures are absorbed by the larger sink droplet on the surface. Can such events induce the roaming effect?

We hope to provide a comprehensive response to both comments in the following.

The work¹ of Yan et al. discusses a condensation phenomenon, hierarchical condensation, on hierarchically structured surfaces. Small condensate droplets form within the microcavities below larger droplets suspended on top of these cavities. Refer to **Figure R4.1** for an illustration recreated based on Ref.¹. For favourable microcavity morphologies, the small droplets can navigate out of the cavity towards the larger droplet. Upon coalescence with the larger droplet, also known as the sink, the small droplet appears to be “absorbed” into it. New condensate droplets can then form within the emptied cavity, and the cycle repeats.¹

Figure R4.1: Hierarchical condensation as described in Ref.¹.

Our response in the following consists of two parts. In Part I, we will show why hierarchical condensation simply cannot occur on boehmite and other solely nanostructured surfaces; thus roaming cannot be induced by it. (Note that in **Supplementary Information S3**, we have already addressed the fundamental intrinsic differences between some of the possible coalescence mechanisms on nanostructured and hierarchical surfaces. We have deliberately selected solely nanostructured surfaces in the main work to exclude any possible effects from microstructures.)

Then, in Part II, we will explain why even if there is hierarchical condensation, it is not possible to induce roaming. Wetted nanocavities are more difficult to dewet. Most importantly, we claim that no substantial tangential momentum can be generated from coalescence of a large droplet with condensate within a cavity, given the size difference.

Part I

Size and morphology of surface structures

Hierarchical condensation may be observed across a large range of droplet sizes, as the reviewer has pointed out. The work¹ considers sink droplets of radii between $\approx 20 \mu\text{m}$ and $500 \mu\text{m}$ (Figure 4b in Ref.¹), and the mean main droplet equivalent radius ($79 \pm 28 \mu\text{m}$) in roaming does lie within this range. However, in hierarchical condensation, these larger droplets are suspended over large microstructures. For example, the microstructure features in Figures 2a to 2f in Ref.¹ has a pitch of $\approx 30 - 40 \mu\text{m}$. In comparison, droplets on boehmite surfaces are suspended over features with characteristic sizes of $\approx 20 - 40 \text{ nm}$, 3 orders of magnitude lower. As a result, the presence of a large number of droplets below a suspended droplet in Ref.¹ is not possible on solely nanostructured surfaces for hierarchical condensation. The nanostructure cavities can only accommodate at most a few nuclei at reasonable subcooling. See **Figure R4.2**.

Figure R4.2: Difference in structure sizes and condensation within the cavities.

Moreover, hierarchical condensation requires the constant inflow of vapour into the structure cavities (v_c in Figures 4a and 4c in Ref.¹, see also **Figure R4.1** above) so that condensate can form below the suspended droplets.¹ This is only possible with open structure cavities, where there are wide paths to allow the flow of vapour. Nanostructures are typically too dense and thin for vapour to flow through. For boehmite, the cavities are also closed on the sides by nanowalls. Therefore, they can only be wetted when directly exposed to the vapour without a droplet on top, since there is no straightforward flow path for vapour to enter below a droplet suspending on top of these structures.

The difference in size and morphology of surface structures between our work and Ref.¹ already suggests that hierarchical condensation cannot occur on nanostructured surfaces, such as boehmite, and therefore cannot be responsible for roaming motion. Although nanocavities could become wetted when they are directly exposed to vapour and the critical nucleation diameter is sufficiently small, they are very unlikely to dewet by coalescing with and appearing to be “absorbed” into the larger droplet above the structures.

Part II

Wettability within the structures

In hierarchical condensation in Ref.¹, the small condensate droplets in the cavities are in the nano-Cassie state, suspended on top of nanostructures which are overlayed onto the microstructures. Therefore, they are sufficiently mobile to spontaneously navigate to the sink droplet due to favourable Laplace pressure difference caused by diverging microcavities. The micro- and nanostructure cavities are not wetted.

On solely nanostructured surfaces in roaming, the nanocavities do not have another tier of roughness and they are hydrophobic instead of superhydrophobic. Condensation within them can result in truly wetted cavities, and dewetting them requires more (and external) work, which is provided by the coalescence of participating droplets during roaming motion. Before roaming, these cavities remain wetted, resulting in partial-Wenzel droplets above. See **Figure R4.3**. These partial-Wenzel droplets do not spontaneously dewet and remain stable in their wetting state if there is no work done.

The concurrent presence of Cassie and (partial-)Wenzel-state condensate droplets on superhydrophobic surfaces has been reported before, as we have included in the manuscript (p. 17):

As some nanocavities fill, stochastic wetting at random locations across the surface promotes the concurrent presence of Cassie and (partial-)Wenzel-state condensate droplets on superhydrophobic surfaces.²

Dewetting of cavities by condensate motion has as well been suggested before for the case of droplet jumping, as we have included in the manuscript (p. 13):

... such stochastic locally wetted cavities are expected to dewet in a roaming event, similar to dewetting by condensate motion previously suggested³ for the case of droplet jumping.

Figure R4.3: Mobility of condensate within the structures.

Most importantly, in any case, the coalescence of two liquid bodies with a vast size mismatch cannot produce enough momentum required for roaming, as we show in the following.

Coalescence with high size mismatch

During coalescence, excess surface energy is released. For this to convert to bulk droplet motion, there has to be a pathway for momentum generation. In conventional droplet jumping, there is a rapidly expanding liquid bridge which impinges on a surface. The surface breaks symmetry of the coalescence oscillations and generate a reaction force for the droplet to travel normally away from the surface.

From the hydrodynamic perspective, the liquid bridge cannot impinge on the surface with a sufficient force (or at all) when there is a high size mismatch. This is also evident in the snapshots of Figure 1d in Ref.¹, in which a 218 μm -diameter droplet coalesces with a 48 μm -diameter droplet, as the authors observe and we quote verbatim from Ref.¹: “*the larger primary droplet remained almost unchanged in both volume and position, resembling a sink*”. In hierarchical condensation of the paper,¹ the droplets in the microstructure cavities are even smaller ($\sim 10 \mu\text{m}$), thus there is an even larger size mismatch. On solely nanostructured surfaces like boehmite in our work, the condensate within the nanostructure cavities is further reduced to the scale of $\sim 10 \text{ nm}$. It is expected that for coalescence between condensate differing in size by orders of magnitude, the smaller body is reduced to becoming a small perturbation to the larger droplet, without any material effect to the coalesced bulk. This perturbation travels along the surface of the coalesced droplet and is eventually viscously dissipated.

From the energetic perspective, the surface energy released from coalescence events decreases as the mismatch increases. Consider a smaller droplet of radius r_1 and a larger droplet of radius $r_2 > r_1$. They coalesce on an ideal non-wetting surface such that they can be considered spherical. They have a total volume V_{12} which is conserved during the coalescence. Then

$$V_{12} = \frac{4}{3}\pi r_1^3 + \frac{4}{3}\pi r_2^3 = \frac{4}{3}\pi r_{12}^3$$

where r_{12} is the radius of the droplet after coalescence. Simplifying the expression, we get

$$r_1^3 + r_2^3 = r_{12}^3 \quad (1)$$

Next, we define the size mismatch between the two droplets as

$$M = \frac{r_2 - r_1}{r_1 + r_2}$$

which is equivalent to the definition based on diameters in **Supplementary Information S12** (p. 61). We rearrange for r_2 and obtain

$$r_2 = r_1 \frac{1 + M}{1 - M} \quad (2)$$

Substituting Equation (2) into Equation (1), we obtain

$$r_1^3 = r_{12}^3 - r_1^3 \left(\frac{1 + M}{1 - M} \right)^3$$

$$r_1^3 = r_{12}^3 \left[1 + \left(\frac{1 + M}{1 - M} \right)^3 \right]^{-1}$$

We obtain an expression for r_1^2 as follows:

$$r_1^2 = r_{12}^2 \left[1 + \left(\frac{1+M}{1-M} \right)^3 \right]^{-2/3} \quad (3)$$

From Equation (2), we have

$$r_2^2 = r_1^2 \left(\frac{1+M}{1-M} \right)^2 \quad (4)$$

Substituting Equation (3) into Equation (4), we get

$$\begin{aligned} r_2^2 &= r_{12}^2 \left[1 + \left(\frac{1+M}{1-M} \right)^3 \right]^{-2/3} \left(\frac{1+M}{1-M} \right)^2 \\ r_2^2 &= r_{12}^2 \left[1 + \left(\frac{1+M}{1-M} \right)^{-3} \right]^{-2/3} \end{aligned} \quad (5)$$

The surface energy is the product between the surface area and surface tension γ . The total surface energy of the two droplets before coalescence is therefore

$$(4\pi r_1^2 + 4\pi r_2^2)\gamma$$

and the surface energy of the droplet after coalescence is

$$4\pi r_{12}^2\gamma$$

We can then compute the excess surface energy E_{ex} that is released from coalescence, normalised by the surface energy before coalescence:

$$\begin{aligned} E_{\text{ex}} &= 1 - \frac{4\pi r_{12}^2\gamma}{(4\pi r_1^2 + 4\pi r_2^2)\gamma} \\ E_{\text{ex}} &= 1 - \frac{r_{12}^2}{r_1^2 + r_2^2} \end{aligned} \quad (6)$$

Substituting Equations (3) and (5) into Equation (6), we get E_{ex} in terms of the mismatch M :

$$\begin{aligned} E_{\text{ex}} &= 1 - \frac{r_{12}^2}{r_{12}^2 \left[1 + \left(\frac{1+M}{1-M} \right)^3 \right]^{-2/3} + r_{12}^2 \left[1 + \left(\frac{1+M}{1-M} \right)^{-3} \right]^{-2/3}} \\ E_{\text{ex}} &= 1 - \left\{ \left[1 + \left(\frac{1+M}{1-M} \right)^3 \right]^{-2/3} + \left[1 + \left(\frac{1+M}{1-M} \right)^{-3} \right]^{-2/3} \right\}^{-1} \end{aligned} \quad (7)$$

Finally, we plot Equation (7) in **Figure R4.4** for $0 \leq M < 1$. It is clear that when mismatch is high, minimal surface energy is released. On solely nanostructured surfaces such as boehmite, even if the condensate within the wetted nanocavities can coalesce with the microdroplets on top of the structures, the vast size mismatch of at least 3 orders of magnitude (e.g. 40 nm vs. 40 μm , $M = 0.998$) releases negligible surface energy $E_{\text{ex}} \approx 0$. The low released surface energy is further compounded with the fact that only a small proportion of released surface energy (if hydrodynamically possible) can be converted to kinetic energy of bulk droplet motion. Roaming of microdroplets is therefore not possible from such coalescence. It then comes as no surprise that no unidirectional in-plane motion has been reported in the work¹ of Yan et al. for hierarchical condensation, since even an order of magnitude size difference (e.g. 10 μm vs. 100 μm) already gives $M = 0.818$ and $E_{\text{ex}} = 0.009$.

Figure R4.4: Normalised excess surface energy E_{ex} for different mismatch M .

In summary, while the work¹ of Yan et al. presents an interesting condensation phenomenon on hierarchically structured superhydrophobic surfaces, we believe it cannot be responsible for roaming motion. Roaming requires a pathway to convert sufficient excess surface energy from coalescence to tangential kinetic energy, in particular through simultaneous presence of droplets in different wetting states. Hierarchical condensation as described in Ref.¹ does not provide such a pathway.

We have modified the manuscript (p. 13) to include the work of Yan et al.¹:

On the other hand, hierarchical condensation,¹ in which condensate nucleates within structure cavities under a droplet, is very unlikely, due to the lack of low-resistance vapour flow paths across the thin, dense and closed nanostructure (**Supplementary Information S3**). After roaming renews the surface, the dewetted cavities are again exposed to the vapour for another nucleation cycle.

In **Supplementary Information S3** (p. 24), we have added a summary of the above discussion:

Hierarchical condensation

On hierarchical surfaces, which possess nanostructures overlayed onto microstructures, hierarchical condensation may occur as described in Ref.¹. It is a phenomenon where small condensate droplets form within the microcavities below larger droplets suspended on top of these cavities (**Figure S3.2**). For favourable microcavity morphologies, the small droplets can navigate out of the cavity towards the larger droplet. Upon coalescence with the larger droplet, also known as the sink, the small droplet appears to be “absorbed” into it. New condensate droplets can then form within the emptied cavity, and the cycle repeats.¹

Figure S3.2: Hierarchical condensation as described in Ref.¹.

Hierarchical condensation cannot occur on boehmite or other solely nanostructured surfaces. In hierarchical condensation, the larger droplets are suspended over large microstructures ($\approx 30 - 40 \mu\text{m}$ in pitch in Ref.¹). In comparison, droplets on boehmite surfaces are suspended over features with characteristic sizes of $\approx 20 - 40 \text{ nm}$, 3 orders of magnitude lower. As a result, the presence of a large number of droplets below a suspended droplet depicted in **Figure S3.2** and Ref.¹ is not possible on solely nanostructured surfaces for hierarchical condensation. The nanostructure cavities can only accommodate at most a few nuclei at reasonable subcooling (**Figure S3.3**).

Figure S3.3: Difference in structure sizes and condensation within the cavities.

Moreover, hierarchical condensation requires the constant inflow of vapour into the structure cavities so that condensate can form below the suspended droplets.¹ This is only possible with open structure cavities, where there are wide paths to allow the flow of vapour. Nanostructures are typically too dense and thin for vapour to flow through. For boehmite and similar nanostructures, the cavities are also closed on the sides by nanowalls. Therefore, they can only be wetted when directly exposed to the vapour from their top opening, that is, without the presence of a droplet at this location blocking the vapour access (**Figure S3.2** and left of **Figure S3.3**). There is no other straightforward flow path for vapour to enter below the droplet suspended on top of these structures blocking the top opening.

In summary, hierarchical condensation cannot occur on boehmite or other solely nanostructured surfaces, and thus cannot be responsible for roaming motion on our surfaces.

It can further be shown that even if there is hierarchical condensation, it is not possible to induce roaming. Firstly, hierarchical condensation on solely nanostructured surfaces would mean wetted nanocavities. In **Figure S3.2**, hierarchical condensation results in condensate within the microstructures in the non-Cassie state, suspended on top of nanostructures with high mobility.¹ The micro- and nanostructure cavities are not wetted. However, on solely nanostructured surfaces, the nanocavities do not have another tier of roughness. Condensation within them can result in truly wetted cavities, and dewetting them requires more (and external) work, which is provided by the coalescence of participating droplets during roaming motion. Before roaming, these cavities remain wetted, resulting in stable partial-Wenzel droplets above as we describe in the main text.

Secondly, the coalescence of two liquid bodies with a vast size mismatch cannot produce enough momentum required for roaming. From the hydrodynamic perspective, the smaller body is reduced to becoming a small perturbation to the larger droplet. This perturbation is viscously dissipated without material effect to the coalesced bulk. From the energetic perspective, up to $\approx 20\%$ of surface energy is released³ upon coalescence when there is no size mismatch between two spherical droplets ($M = (r_2 - r_1)/(r_1 + r_2) = 0$, r_1 and r_2 are the radii of a smaller and larger droplet respectively). When there is an order of magnitude size difference¹ (e.g. $10\ \mu\text{m}$ vs. $100\ \mu\text{m}$), $M = 0.818$ and only 0.9% of surface energy is released. On solely nanostructured surfaces, the difference is even larger and the released surface energy becomes negligible. The calculations are shown below.

Consider two droplets of radii r_1 and $r_2 > r_1$. They coalesce on an ideal non-wetting surface such that they can be considered spherical. The total volume V_{12} is conserved during coalescence:

$$V_{12} = \frac{4}{3}\pi r_1^3 + \frac{4}{3}\pi r_2^3 = \frac{4}{3}\pi r_{12}^3$$

where r_{12} is the radius of the droplet after coalescence. Then,

$$r_1^3 + r_2^3 = r_{12}^3$$

For $M = (r_2 - r_1)/(r_1 + r_2)$ defined as the size mismatch (equivalent to the definition based on diameters in **Supplementary Information S12**),

$$r_2 = r_1 \frac{1 + M}{1 - M}$$

Substituting the equations, we have the following expressions for r_1^2 and r_2^2 :

$$r_1^2 = r_{12}^2 \left[1 + \left(\frac{1 + M}{1 - M} \right)^3 \right]^{-2/3}$$

$$r_2^2 = r_{12}^2 \left[1 + \left(\frac{1 + M}{1 - M} \right)^{-3} \right]^{-2/3}$$

The surface energy is the product between the surface area and surface tension γ . The total surface energy of the two droplets before coalescence is

$$(4\pi r_1^2 + 4\pi r_2^2)\gamma$$

and the surface energy of the droplet after coalescence is

$$4\pi r_{12}^2 \gamma$$

The excess surface energy E_{ex} that is released from coalescence, normalised by the surface energy before coalescence, is

$$E_{\text{ex}} = 1 - \frac{4\pi r_{12}^2 \gamma}{(4\pi r_1^2 + 4\pi r_2^2) \gamma} = 1 - \left\{ \left[1 + \left(\frac{1+M}{1-M} \right)^3 \right]^{-2/3} + \left[1 + \left(\frac{1+M}{1-M} \right)^{-3} \right]^{-2/3} \right\}^{-1}$$

Figure S3.4 plots E_{ex} for $0 \leq M < 1$, where E_{ex} approaches 0 as M approaches 1.

Figure S3.4: Normalised excess surface energy E_{ex} for different mismatch M .

In conclusion, not only hierarchical condensation, as described in Ref¹, cannot occur on solely nanostructured surfaces, but even if it were to occur (hypothetically), it would not provide a viable pathway for momentum generation. Roaming cannot result from hierarchical condensation.

REFERENCES

1. Yan, X. *et al.* Hierarchical Condensation. *ACS Nano* **13**, 8169–8184 (2019).
2. Miljkovic, N., Enright, R. & Wang, E. N. Effect of Droplet Morphology on Growth Dynamics and Heat Transfer during Condensation on Superhydrophobic Nanostructured Surfaces. *ACS Nano* **6**, 1776–1785 (2012).
3. Boreyko, J. B. & Chen, C.-H. Self-Propelled Dropwise Condensate on Superhydrophobic Surfaces. *Phys. Rev. Lett.* **103**, 184501 (2009).